# Secant Line Search for Frank-Wolfe Algorithms

**Deborah Hendrych** [* 1 2] **Sebastian Pokutta** [* 1 2] **Mathieu Besançon** [* 3] **David Martínez-Rubio** [* 4]

## Abstract

We present a new step-size strategy based on the secant method for Frank-Wolfe algorithms. This strategy, which requires mild assumptions about the function under consideration, can be applied to any Frank-Wolfe algorithm. It is as effective as full line search and, in particular, allows for adapting to the local smoothness of the function, such as in Pedregosa et al. (2020), but comes with a significantly reduced computational cost, leading to higher effective rates of convergence. We provide theoretical guarantees and demonstrate the effectiveness of the strategy through numerical experiments.

## 1. Introduction

We are interested in solving constrained optimization problems of the form

$$\min_{\mathbf{x} \in \mathcal{X}} f(\mathbf{x}) \tag{1.1}$$

with a first-order method, relying only on gradient and function evaluations, where $f$ is a smooth function and $\mathcal{X}$ is a compact convex set onto which projection is potentially expensive. In this case, variants of the Frank-Wolfe (FW) algorithm (Frank & Wolfe, 1956; Levitin & Polyak, 1966) are a popular choice. One operation required for most FW variants is a choice of a step size to update the iterate $\mathbf{x}_t$ along a descent direction. We present a new line search strategy based on the secant method for these algorithms. To this end, we will solve for roots of the optimality system of the line search problem. Together with the first-order requirement of the main algorithm, this means that only

subproblem's function evaluations are available to us, which corresponds to computing partial derivatives of $f$, making the secant method (instead of the Newton-Raphson method) the natural choice.

**Related Work**

There is an abundance of work on line search strategies for optimization, as step-size rules can significantly impact the performance of optimization algorithms, both computationally and in terms of convergence rates. Standard examples of step-size strategies in unconstrained optimization include backtracking and golden ratio search (see e.g., (Nocedal & Wright, 1999)). A notable recent example is the Silver Step-size Schedule (Altschuler & Parrilo, 2023; 2024a;b), which achieves partial acceleration for smooth convex optimization via standard gradient descent. Here, we will only provide a brief overview of the prior works most closely related to ours.

The requirements for a step-size strategy are typically that it should be (1) effective (making progress), (2) efficient (low computational cost), and (3) adaptive (adapting to the local smoothness of the function). As a bonus, the strategy should be simple and robust so that it is easy to implement. While achieving two out of the three requirements is relatively easy, achieving all three is challenging. Recently, Malitsky & Mishchenko (2020; 2023) made significant progress in this direction by providing an adaptive step-size strategy that does not perform line search, satisfying all three requirements. This sparked a lot of interest, with alternative approaches (see e.g., Zhou et al. (2024) based on the Barzilai-Borwein two-point step-size (Barzilai & Borwein, 1988)) being analyzed. These approaches estimate the inverse of the local Lipschitz smoothness constant via two points $\mathbf{x}, \mathbf{y}$ as $\frac{\|\mathbf{x}-\mathbf{y}\|}{\|\nabla f(\mathbf{x})-\nabla f(\mathbf{y})\|}$ and then use this (with modifications) as the step-size $\gamma_t$ for an update of the form $\mathbf{x}_{t+1} \leftarrow \mathbf{x}_t - \gamma_t \nabla f(\mathbf{x}_t)$. A close inspection reveals that this is effectively a modified (higher-dimensional) secant step. The resulting step-size strategy performs well computationally with great adaptivity to the function geometry. Unfortunately, the approach comes with two caveats for the setting we are interested in. First, it cannot be carried over to the Frank-Wolfe setting as the analysis relies on the convergence of the iterates to the optimal point. This, however, is not necessarily given for Frank-Wolfe algorithms (Bolte

---

[*]Equal contribution  [1]Zuse Institute Berlin, Berlin, Germany [2]Berlin Institute of Technology, Berlin, Germany [3]Université Grenoble Alpes, Inria, Laboratoire d'Informatique de Grenoble, Grenoble, France [4]Signal Theory and Communications Department, Carlos III University of Madrid, Madrid, Spain. Correspondence to: Sebastian Pokutta <pokutta@zib.de>, Mathieu Besançon <mathieu.besancon@inria.fr>, David Martínez-Rubio <dm-rubio@ing.uc3m.es>.

*Proceedings of the 42^{nd} International Conference on Machine Learning*, Vancouver, Canada. PMLR 267, 2025. Copyright 2025 by the author(s).

et al., 2024), due to their affine-invariance. We point out that this is also not just an artifact but the proposed step-size strategy deliberately gives up the monotonic descent requirement to achieve larger steps, which is incompatible with Frank-Wolfe convergence analysis. Second, the methods use dampened secant steps which lead to the loss of superlinear convergence of the secant method as can be seen from Equation (2.6) in Section 2.2, where the equality does not hold anymore as the lower-order terms do not cancel. This is not an issue in the settings considered by the aforementioned works as they perform single secant steps only. However, we want to capitalize on the potentially super-linear convergence.

Step-size strategies for Frank-Wolfe can essentially be grouped into three types (see e.g., (Braun et al., 2022) for an extensive overview): (1) open-loop strategies of the form $\gamma_t = \frac{\ell}{\ell+t}$ with $\ell \geq 2$ being an integer. These strategies recently received new attention as they can achieve higher convergence rates than e.g., line search strategies in some settings (see e.g., (Bach, 2021; Wirth et al., 2023; 2024)) (ii) so-called short-steps, which are effectively the solution arising from the smoothness inequality but require knowledge of the gradient Lipschitz constant (iii) line search strategies, where basically any strategy can be used, however, the current gold standard is the adaptive backtracking line search of Pedregosa et al. (2020), which also approximates the local smoothness constant and its numerically more stable variant from Pokutta (2024). Another recent successful approach has been the monotonic open-loop strategy of (Carderera et al., 2021; 2024), which was originally developed for (generalized) self-concordant functions but often performs very well in general. We highlight that even though we gathered these methods under "line search", they are strictly speaking not line search methods since they terminate with conditions other than (near-)optimality on the segment given by the descent direction. The adaptive step sizes from Pedregosa et al. (2020); Pokutta (2024) test that a local Lipschitz smoothness estimate produces a valid lower bound on the primal progress while the monotonic step size from Carderera et al. (2021; 2024) tests primal improvement on top of an open-loop step-size schedule.

Other notable recent approaches in the context of unconstrained optimization include speeding up the backtracking search with an adaptive retraction (Cavalcanti et al., 2024), which can be applied broadly and might be adaptable to the Frank-Wolfe setting. Moreover, the local convergence of quasi-Newton methods of the Broyden class have been heavily studied in recent work (see e.g., (Jin & Mokhtari, 2023; Rodomanov & Nesterov, 2022; 2021)) to provide finer and finite-time convergence guarantees. While highly interesting in their own right, these approaches are only tangentially related to our work here as we directly use the secant method and many of the challenges of quasi-Newton methods do not apply here. Stabilization of locally convergent algorithms has been a topic that already received significant attention decades ago, see e.g., (Polak, 1975; 1976) and the secant method has been also used for the solution of simultaneous nonlinear equations (Wolfe, 1959).

**Contribution**

Our contribution can be summarized as follows:

**New step-size strategy.** With line searches, the key trade-off is between the cost of the search, which reduces the effective rate of convergence, and improved step sizes. We provide a new step-size strategy, coined *Secant Line Search (SLS)*, which solves the line search problem via the secant method. This leads to an extremely fast strategy and, while it technically does not meet all previously-listed requirements for a step size due to its search-type nature, the iteration count is usually so low (around $6 - 7$ iterations) that it practically does satisfy all requirements. Moreover, the strategy is adaptive in the sense that it exploits favorable local smoothness and it can be applied to any Frank-Wolfe algorithm, due to the constrained nature of the problems Frank-Wolfe is applied to. In fact, for quadratics, SLS converges in one iteration and exploits the local smoothness of the function in the direction of the descent.

**Theoretical guarantees.** Using the secant method to solve the line search problem comes with multiple issues for most optimization algorithms since the secant method, while often very fast, is not necessarily convergent. However, due to the special structure of FW algorithms that do not follow the gradient but use alternative directions of the form $\mathbf{x}_t - \mathbf{v}_t$, cf. Algorithm 1, the solution to the line search problem is confined to a bounded interval $[0, \gamma_{\max}]$. Exploiting this property, we can guarantee convergence under mild assumptions in Lemma 2.1 and Theorem 3.1.

**Numerical experiments.** We provide extensive numerical experiments demonstrating the superior computational performance of SLS over other step sizes. We also discuss implementation enhancements that further reduce computational costs, making SLS a practical option for a broad class of problems, being either the best performing option or close to the best one. The results show that SLS not only accelerates the convergence of FW compared to other step-size strategies but is also highly competitive in time thanks to a low number of secant iterations within the line search. This low line search cost and improved convergence make SLS an excellent new step size choice for FW algorithms for a broad range of problems. Our strategy might be more broadly applicable to other algorithm classes beyond FW but may require stronger assumptions or safeguards to guarantee convergence.

**Preliminaries and Notation**

We use bold-faced letters $\mathbf{x}$ to denote vectors and non-bold-faced letters $x$ to denote scalars. Throughout the paper, we will use $t$ for the iteration count of the outer algorithm, typically iterations of the FW method and $n$ for the iteration count of the line search method. We let $\|\cdot\|$ denote the Euclidean norm and $\langle \cdot, \cdot \rangle$ the standard inner product.

## 2. The Secant Method

The Secant Method (see e.g., Papakonstantinou & Tapia (2013)) is a well-known method for finding the root of a function $\varphi : \mathbb{R} \to \mathbb{R}$. It is based on the recursion:

$$x_{n+1} \leftarrow x_n - \varphi(x_n) \cdot \frac{x_n - x_{n-1}}{\varphi(x_n) - \varphi(x_{n-1})}, \qquad (2.1)$$

with initial values $x_0, x_1$ chosen appropriately. We can think of the secant method as an approximate version of Newton's method, which uses the update

$$x_{n+1} \leftarrow x_n - \varphi(x_n) \cdot \frac{1}{\varphi'(x_n)} \qquad (2.2)$$

using the approximation:

$$\varphi'(x_n) \approx \frac{\varphi(x_n) - \varphi(x_{n-1})}{x_n - x_{n-1}}. \qquad (2.3)$$

The secant method predates Newton's method by over 3000 years (Papakonstantinou & Tapia, 2013).

One should think about $\varphi(\gamma) = f(x_t - \gamma \nabla f(x_t))$ in our application. Before considering the line search problem, we establish global convergence when the absolute value of the derivative of the function $\varphi$ is increasing as we move away from the root, e.g., if $\varphi'' \varphi > 0$. We also recall the convergence rate of the secant method near the root. The latter is folklore and many different proofs exist, we provide a sketch of the argument only. For global convergence, the proof is probably known (and straightforward) but we are not aware of any direct reference and thus we provide it here. It ensures convergence under suitable initialization and mild assumptions for our setting. Throughout this section we assume that $\varphi$ is a smooth function and sufficiently differentiable.

### 2.1. Global Convergence

In general, the secant method is not globally convergent. However, we can show that under suitable assumptions, the secant method converges monotonically to a root in cases relevant for our line search problem. To simplify the exposition, we introduce the notation $\Delta(x, y) \doteq \frac{\varphi(x) - \varphi(y)}{x - y}$ and $S(x, y) \doteq x - \frac{\varphi(x)}{\Delta(x,y)}$, so that the secant method can be written as:

$$x_{n+1} \leftarrow S(x_n, x_{n-1}) = x_n - \frac{\varphi(x_n)}{\Delta(x_n, x_{n-1})}. \qquad (2.4)$$

Clearly $\Delta(x, y) = \Delta(y, x)$. Further, observe that $S(x, y) = x - \frac{\varphi(x)}{\Delta(x,y)} = y - \frac{\varphi(y)}{\Delta(x,y)} = S(y, x)$. We will now establish a first convergence result of the secant method.

**Lemma 2.1.** *Let $a$ be a root of $\varphi : \mathbb{R} \to \mathbb{R}$ and let $\mathcal{U}$ be a one-sided neighborhood of $a$. Further assume that $|\varphi'(x)|$ is strictly increasing in $|x - a|$ with $x \in \mathcal{U}$, i.e., as we move away from $a$; in particular $a$ is the only root on $\mathcal{U}$ and $\varphi$ is monotone on $\mathcal{U}$. Then, we have:*

$$0 < \frac{S(x, y) - a}{x - a} < 1,$$

*for all $a \neq x, y \in \mathcal{U}$, i.e., the distance to $a$ is strictly decreasing.*

*Proof.* Note that $\varphi(a) = 0$ and let $x, y \in \mathcal{U}$ be arbitrary with $x, y, a$ all distinct. We differentiate the following two cases.

*Case 1:* $|x - a| < |y - a|$, i.e., $x$ is closer to $a$. Observe that we have:

$$\frac{S(x, y) - a}{x - a} = \frac{x - \frac{\varphi(x)}{\Delta(x,y)} - a}{x - a}$$
$$= 1 - \frac{\varphi(x) - \varphi(a)}{\Delta(x,y)(x - a)} = 1 - \frac{\Delta(x, a)}{\Delta(x, y)}.$$

By monotonicity $\frac{\Delta(x,a)}{\Delta(x,y)} > 0$. Moreover, since $|\varphi'(x)|$ is stricly increasing as we move away from $a$ in $\mathcal{U}$, we have $\Delta(x, a) < \Delta(x, y)$, so that $0 < \frac{S(x,y)-a}{x-a} < 1$ follows; this also implies that we do not leave $\mathcal{U}$ and stay on the same side of $a$ as $\frac{S(x,y)-a}{x-a} > 0$

*Case 2:* $|x - a| > |y - a|$, i.e., $y$ is closer to $a$. Similar as before after factoring out $\frac{y-a}{x-a}$ and using $S(x, y) = S(y, x)$ we similarly obtain:

$$\frac{S(x, y) - a}{x - a} = \frac{y - a}{x - a} \cdot \frac{y - \frac{\varphi(y)}{\Delta(x,y)} - a}{y - a}$$
$$= \frac{y - a}{x - a} \left(1 - \frac{\Delta(y, a)}{\Delta(x, y)}\right).$$

Now observe that $0 < \frac{y-a}{x-a} < 1$ and $\Delta(y, a) < \Delta(x, y)$ as above, so that $0 < \frac{S(x,y)-a}{x-a} < 1$ follows.

Now, the property $0 < \frac{S(x,y)-a}{x-a} < 1$ proven in the two cases above imply that $x_{n+1} - x_n$ converges to 0. Moreover, $x_n - x_{n+1} = \frac{\varphi(x_n)}{\Delta(x_n, x_{n-1})}$ and $|\Delta(x_n, x_{n-1})| < \max\{|\varphi'(x_0)|, |\varphi'(x_1)|\}$, which implies that $\varphi(x_n)$ converges to 0 and thus $x_n$ converges to $a$. $\qquad \square$

The lemma above basically states that if, in a neighborhood $\mathcal{U}$ of the root, the function is monotone, has strict curvature, and the initial points $x_0, x_1$ are both on the same side

of the root, then the secant method converges monotonically to the root. This is for example the case if $\varphi'' \varphi > 0$ holds. We will later see that in case of our line search problem, apart from the strict curvature condition, all other conditions are satisfied naturally. Note, that the requirement that $|\varphi'(x)|$ is strictly increasing can be relaxed to require $\Delta(a, x) < \Delta(x, y)$ whenever $x$ lies between $a$ and $y$ and $\Delta(a, x), \Delta(x, y)$ having the same sign, not requiring that $f$ to be smooth.

**Remark 2.2** (Rate and Order of Convergence). Note that Lemma 2.1 makes no statement about the order of convergence. This is for good reason as arbitrarily slow convergence (still linear though with small rates) is compatible with the Lemma 2.1 both globally and locally. In fact, the convergence order and rate is a function of the multiplicity of the root. As a simple example consider $\varphi(x) = x^m$ with $m > 2$, which satisfies our assumptions. The convergence rate of the secant method is linear with rate $\lambda$ with $0 < \lambda < 1$, so that $\lambda^m + \lambda^{m-1} = 1$; see e.g., (Díez, 2003). Note, though that if the secant method converges it converges at least with order 1, i.e., linearly.

## 2.2. Local Convergence

Let $\varphi(a) = 0$. For the sake of exposition we consider the case of a simple root and assume that $\varphi$ is twice differentiable with $\varphi'(a) \neq 0$ and $\varphi''(a) \neq 0$. We define the iterates as $x_n = a + \epsilon_n$ to perform an error analysis with the recursion:

$$\epsilon_{n+1} = \epsilon_n - \varphi(a+\epsilon_n) \cdot \frac{\epsilon_n - \epsilon_{n-1}}{\varphi(a + \epsilon_n) - \varphi(a + \epsilon_{n-1})}. \quad (2.5)$$

Using a Taylor expansion:

$$\varphi(a + \epsilon) \approx \varphi'(a)\epsilon + \frac{\varphi''(a)}{2}\epsilon^2 = \epsilon\varphi'(a)(1 + M\epsilon), \quad (2.6)$$

where $M \doteq \frac{\varphi''(a)}{2\varphi'(a)}$, we can derive the recursion:

$$\epsilon_{n+1} \approx \frac{\varphi''(a)}{2\varphi'(a)}\epsilon_{n-1}\epsilon_n. \quad (2.7)$$

By analyzing the series of $\log(\epsilon_n)$ (Díez, 2003), we can derive an error estimate of:

$$|x_{n+1} - a| \approx \left| \frac{\varphi''(a)}{2\varphi'(a)} \right|^{\frac{\sqrt{5}-1}{2}} |x_n - a|^{\frac{1+\sqrt{5}}{2}}, \quad (2.8)$$

i.e., we have a convergence of order $\frac{1+\sqrt{5}}{2} \approx 1.618$. For a full proof, see e.g., Grinshpan (2024) and for the convergence for roots of higher order we refer the reader to Díez (2003). In fact, for roots of order $m \geq 2$, the convergence rate drops from superlinear to linear with rate $\lambda$ with $0 < \lambda < 1$, so that $\lambda^m + \lambda^{m-1} = 1$; see e.g., Díez (2003).

**Remark 2.3** (Secant Method vs Newton's Method in Black-Box Optimization). The secant method has an advantage over Newton's method in black-box optimization, where only function evaluations are available, as it avoids gradient computations and requires just one function evaluation per iteration. However, even when both function and gradient evaluations are available in an oracle model at roughly the same cost, we can perform roughly two secant iterations for every Newton iteration, potentially making the secant method faster in practice, despite its lower order of convergence (order $\approx 1.618$) compared to Newton's quadratic convergence (order 2) with effective orders of convergence of $1.618^2 \approx 2.62$ vs. 2.

In white-box settings, where gradients can be computed efficiently, such as through reverse automatic differentiation (AD), the function evaluation becomes essentially free as a byproduct of the gradient evaluation, making Newton's method potentially more favorable due to its faster convergence rate. We would like to stress that essentially all our arguments also apply to a variant that would use Newton's method for the line search problem; in Lemma 2.1 we would simply invoke the mean value theorem and replace $\Delta(x, y)$ with $\varphi'(x)$ and then the proof would work analogously.

For our setting however, we assume that we ultimately want to optimize a function $f$ with a first-order oracle and we will run the secant method for the line search problem over the directional derivative of $f$, so that second-order information is not available. For the sake of completeness however, we did run some synthetic tests, see Section B, comparing the Newton's method and the secant method on problems as they appear in our setting, with both AD and user-provided derivatives for the second-order access, and found that in practice the secant method is superior in the first case and almost always faster in the second case. One reason for this is that both in the case of automatic differentiation and manual gradients, each line search subproblem requires a Hessian-vector computation, while not exploiting that information beyond the step size. This "partially second-order" method was already observed in Carderera et al. (2021) not to perform well against a fully first-order method due to the high cost per iteration and little additional gain.

Finally, a note on Brent's method (Brent, 1971), also called Brent-Dekker method, is in order; see also Flannery et al. (1992); Brent (2013) for more details. While this root-finding method is often considered the gold standard for derivative-free root finding with an order of convergence of $\approx 1.839$, it is significantly more involved and expensive than the secant method and in our tests for our problems, was outperformed by the secant method. This can be attributed to the secant method's lower computational cost per iteration and the fact that we do (almost) never require the additional safeguards of Brent's method because of the already present

step size bounds for FW together with Lemma 2.1. We benchmarked in Julia via `Roots.jl` as well as in Python via `SciPy`.

# 3. The Secant Method for Line Search in Frank-Wolfe Algorithms

To disambiguate, we denote the iterates of the secant method applied to the line search problem by $\gamma_n$ and the iterates of the Frank-Wolfe algorithm by $x_t$. We are interested in solving Problem (1.1) with $f$ being a smooth and convex function. In Algorithm 1 we recall the vanilla Frank-Wolfe algorithm and refer the interested reader to Braun et al. (2022) for more advanced variants.

---

**Algorithm 1** Frank-Wolfe algorithm (Frank & Wolfe, 1956; Levitin & Polyak, 1966)

---

1: **Input:** Initial point $\mathbf{x}_0 \in \mathcal{X}$, number of iterations $T$
2: **Output:** Final iterate $\mathbf{x}_T$
3: **for** $t = 0$ to $T - 1$ **do**
4:    $\mathbf{v}_t \leftarrow \underset{\mathbf{v} \in \mathcal{X}}{\operatorname{argmin}} \langle \nabla f(\mathbf{x}_t), \mathbf{v} \rangle$
5:    $\mathbf{x}_{t+1} \leftarrow (1 - \gamma_t)\mathbf{x}_t + \gamma_t \mathbf{v}_t$
6: **end for**

---

Most Frank-Wolfe variants update the iterate as:

$$\mathbf{x}_{t+1} \leftarrow \mathbf{x}_t - \gamma \mathbf{d}_t, \tag{3.1}$$

with $\mathbf{d}_t$ being a direction and $\gamma \in [0, \gamma_{\max}]$ determined by the specific FW variant under consideration. For instance, $\mathbf{d}_t = \mathbf{x}_t - \mathbf{v}_t$ and $\gamma_{\max} = 1$ for a standard FW step as we can see in Algorithm 1. A key point common to all variants is that iterates are maintained as convex combinations of extreme points of the feasible set $\mathcal{X}$, so that typically $\gamma_{\max} \leq 1$.

When the step size $\gamma$ is determined by line search, the goal is to choose $\gamma$ so that progress is maximized, i.e., we solve the line search problem:

$$\gamma \leftarrow \underset{\gamma \in [0, \gamma_{\max}]}{\operatorname{argmin}} f(\mathbf{x}_t - \gamma \mathbf{d}_t). \tag{3.2}$$

This is equivalent to finding the root of the optimality condition:

$$\frac{\partial}{\partial \gamma} f(\mathbf{x}_t - \gamma \mathbf{d}_t) = 0, \tag{3.3}$$

and since we consider convex and smooth functions $f$ such a solution exists. Moreover, we can deliberately ignore the constraint $\gamma \in [0, \gamma_{\max}]$ because if the optimal $\gamma$ is outside of this interval, we can simply clip it to the boundary, which will be the optimal solution to the constrained problem.

Applying the secant method to the line search problem (3.3),

we have $\varphi(\gamma) = \langle f(\mathbf{x}_t - \gamma \mathbf{d}_t), \mathbf{d}_t \rangle$ and use the recursion:

$$\gamma_{n+1} \leftarrow \gamma_n - \langle \nabla f(\mathbf{x}_t - \gamma_n \mathbf{d}_t), \mathbf{d}_t \rangle \cdot$$
$$\frac{\gamma_n - \gamma_{n-1}}{\langle \nabla f(\mathbf{x}_t - \gamma_n \mathbf{d}_t), \mathbf{d}_t \rangle - \langle \nabla f(\mathbf{x}_t - \gamma_{n-1} \mathbf{d}_t), \mathbf{d}_t \rangle}. \tag{3.4}$$

It is important to note that SLS does not approximate the inverse of the local Lipschitz constant of $f$ in contrast to (Malitsky & Mishchenko, 2020; 2023). Rather, it approximates the relevant quantity in the Frank-Wolfe context, which is related (and identical for quadratics) to $\frac{1}{L} \frac{\langle \nabla f(\mathbf{x}_t), \mathbf{d}_t \rangle}{\|\mathbf{d}_t\|^2}$.

An implementation of *Secant Line Search (SLS)* is given in Algorithm 2. The algorithm is purposefully written in a verbose fashion to highlight that, apart from the initial two points, only one gradient evaluation is required per iteration. In Section A, we specify how the convergence rates of Frank-Wolfe behave depending on different choices of the error parameter $\epsilon$ within the line search subproblems.

---

**Algorithm 2** Secant Line Search (SLS)

---

1: **Input:** Function $f$, initial point $\mathbf{x}_t$, direction $\mathbf{d}_t$, initial step sizes $\gamma_0, \gamma_1$, tolerance $\epsilon$
2: **Output:** Step size $\gamma^*$
3: Initialize $\gamma_{-1} \leftarrow \gamma_0, \gamma_0 \leftarrow \gamma_1$
4: Compute $\varphi_{-1} \leftarrow \langle \nabla f(\mathbf{x}_t - \gamma_{-1} \mathbf{d}_t), \mathbf{d}_t \rangle$
5: Compute $\varphi_0 \leftarrow \langle \nabla f(\mathbf{x}_t - \gamma_0 \mathbf{d}_t), \mathbf{d}_t \rangle$
6: **repeat**
7:    Update $\gamma_1 \leftarrow \gamma_0 - \varphi_0 \cdot \frac{\gamma_0 - \gamma_{-1}}{\varphi_0 - \varphi_{-1}}$
8:    $\gamma_1 \leftarrow \max\{0, \min\{\gamma_1, \gamma_{\max}\}\}$    {clip}
9:    Update $\gamma_{-1} \leftarrow \gamma_0, \gamma_0 \leftarrow \gamma_1$
10:    Update $\varphi_{-1} \leftarrow \varphi_0$
11:    Compute $\varphi_0 \leftarrow \langle \nabla f(\mathbf{x}_t - \gamma_0 \mathbf{d}_t), \mathbf{d}_t \rangle$
12: **until** $|\varphi_0| < \epsilon$
13: **return** $\gamma^* \leftarrow \gamma_0$

---

As already indicated earlier we have that $\gamma_a \in [0, 1]$ as the iterates are convex combinations of the extreme points of the feasible set $\mathcal{X}$ with the current iterate. In fact, due to not following the gradient but rather updating $\mathbf{x}_{t+1} \leftarrow (1 - \gamma_t)\mathbf{x}_t + \gamma_t \mathbf{v}_t$ after the first iteration, the optimal solution $\gamma_a$ of (3.3) is almost always in the interval $(0, 1)$. To see that $\gamma_a > 0$, observe that the Frank-Wolfe gap $\langle \nabla f(\mathbf{x}_t), \mathbf{x}_t - \mathbf{v}_t \rangle > 0$ by convexity of $f$ and definition of $\mathbf{v}_t$ as long as $\mathbf{x}_t$ is not optimal and hence $\mathbf{x}_t - \mathbf{v}_t$ is a descent direction. The upper bound typically depends on the function class under consideration. For the upper bound in the case of $L$-smooth functions, see, e.g., (Braun et al., 2022, Remark 2.5) and for (generalized) self-concordant functions, see (Carderera et al., 2021; 2024). We highlight that $\gamma_a < 1$ is not necessary for our arguments to work but it is useful as it basically ensures that the root is to be found in the interval $(0, 1)$. When this is not the case and clipping

occurs, we still make significant progress (at least as much as the optimal short step) and typically such a step halves the primal gap.

With this, we can now establish the convergence of the Secant Line Search (SLS) when used in a Frank-Wolfe algorithm.

**Theorem 3.1** (Convergence of SLS). *Let $f$ be a strictly convex and smooth function and let $\mathcal{X}$ be a compact convex set. Then running the Frank-Wolfe algorithm (Algorithm 1) with SLS initialized with $\gamma_0 = 0$ and $\gamma_1 = \gamma_0 + \rho$, where $\rho$ is a small positive perturbation, converges at its optimal rate (which depends on $f$ and $\mathcal{X}$) and each SLS call converges.*

*Proof.* If SLS converges, then it returns the optimal solution to the line search problem (3.3) and hence the iterates of the Frank-Wolfe algorithm are chosen as if run with exact line search (up to the line search tolerance $\epsilon$), so that the first part of the theorem is trivially true.

It remains to verify the assumptions of Lemma 2.1. As $f$ is strictly convex, so is $f(\mathbf{x}_t - \gamma\mathbf{d}_t)$ for any $\mathbf{x}_t \in \mathcal{X}$ and $\mathbf{d}_t = \mathbf{x}_t - \mathbf{v}_t$ as obtained via Algorithm 1. Hence the derivative $\frac{\partial}{\partial\gamma}f(\mathbf{x}_t - \gamma\mathbf{d}_t)$ of $f(\mathbf{x}_t - \gamma\mathbf{d}_t)$ is strictly increasing in $\gamma$. Moreover, the initial points $\gamma_0 = 0$ and $\gamma_1 = \gamma_0 + \rho$ are on the same side of the root, so that the assumptions of Lemma 2.1 are satisfied. $\square$

While the above theorem only guarantees (linear) convergence of SLS in the line search problems, the following remarks give indications for cases in which we can expect superlinear convergence of SLS.

**Remark 3.2** (Superlinear convergence for self-concordant functions). Let $f$ be self-concordant on $\mathcal{X}$. Then for any $\mathbf{x}_t \in \mathcal{X}$ and $\mathbf{d}_t = \mathbf{x}_t - \mathbf{v}_t$ we have that

$$\left|\frac{\partial^3}{\partial\gamma^3}f(\mathbf{x}_t - \gamma\mathbf{d}_t)(\tau)\right| \leq 2\frac{\partial^2}{\partial\gamma^2}f(\mathbf{x}_t - \gamma\mathbf{d}_t)(\tau)^{\frac{3}{2}} \quad (3.5)$$

for all $\tau \in [0, 1]$. Thus if $\frac{\partial^3}{\partial\gamma^3}f(\mathbf{x}_t - \gamma\mathbf{d}_t)(\tau) \neq 0$ for all $\tau \in [0, 1]$, then $\gamma_a$ is a simple root of Equation (3.3), so that we can expect superlinear convergence.

Moreover, the constant $M$ appearing in Equation (2.6) is obtained as:

$$M = \frac{\frac{\partial^3}{\partial\gamma^3}f(\mathbf{x}_t - \gamma\mathbf{d}_t)(\gamma_a)}{2\frac{\partial^2}{\partial\gamma^2}f(\mathbf{x}_t - \gamma\mathbf{d}_t)(\gamma_a)}. \quad (3.6)$$

In particular in the case of superlinear convergence, SLS achieves very high precision, which is often helpful for ill-conditioned problems, with very few iterations.

**Remark 3.3** (Other FW variants). While we considered the vanilla Frank-Wolfe algorithm, the same analysis can

be applied to basically any other variants of the Frank-Wolfe algorithm, e.g., the Away-Step Frank-Wolfe algorithm (Wolfe, 1970; Lacoste-Julien & Jaggi, 2015), the Pairwise Conditional Gradients (Lacoste-Julien & Jaggi, 2015), the Blended (Pairwise) Conditional Gradients algorithms (Braun et al., 2019; Tsuji et al., 2022), and Decomposition-invariant Conditional Gradients (Garber & Meshi, 2016; Bashiri & Zhang, 2017). Some of these methods – typically those that explicitly maintain a so-called active set, i.e., a subset of extreme points from which a convex combination forms the current iterate – have "drop steps", i.e., iterations that do not guarantee progress but instead sparsify the active set, via e.g., away or pairwise steps when clipping occurs. Nonetheless, the same analysis can be applied to these methods as well by carrying out over the. We present computational experiments for these algorithms as well below and in the appendix.

**Remark 3.4** (Secant Line Search for Quadratics). If $f$ is a convex quadratic, then the line search problem (3.3) is an affine-linear equation. If the iterate $\mathbf{x}_t$ is not optimal yet, then $\mathbf{x}_t - \mathbf{v}_t$ is a descent direction, so that the slope of the affine-linear function is negative and the secant method converges in a single iteration.

As such SLS is almost as cheap as the short-step step size, which would evaluate

$$\gamma = \max\left\{0, \min\left\{\frac{\langle\nabla f(\mathbf{x}_t), \mathbf{x}_t - \mathbf{v}_t\rangle}{L\|\mathbf{x}_t - \mathbf{v}_t\|^2}, 1\right\}\right\},$$

while not requiring knowledge of $L$ and exploiting local smoothness, with progress at least as good as the short step.

For the sake of completeness, note that if we would run Newton's method here, provided we would have access to second-order information, then we would need at least one additional second-order evaluation; which is precisely also what we observed in Section B, where we compared the secant and Newton's method.

## 4. Computational Experiments

We evaluate the performance of SLS on several problem classes. All experiments are carried out in Julia with the blended pairwise FW variant implemented in FrankWolfe.jl (Besançon et al., 2022; 2025) with default parameters and a one-hour time limit. Additional experiments on standard FW are presented in the appendix. An instance is considered solved if the FW gap reaches $10^{-7}$. The implementation of the experiments was done with the Julia package FrankWolfe.jl and is available as part of the package.

**Remark 4.1** (Implementation Details and Improvements). In the following, we discuss some algorithmic considerations ensuring good performance in a wide range of problems.

*Implementation Safeguards.* In the theorem above, we assumed that the function $f$ is strictly convex. This assumption is not necessarily satisfied although the function may be strictly convex on the line we search on. To make SLS robust and safe, whenever the secant method fails, one may run any fallback strategy, e.g., backtracking line search.

*Warm-starting.* We can warm-start SLS by using the optimal $\gamma^*$ from the last call and initializing the secant method with $\gamma_0 = \gamma^*$ and $\gamma_1 = \gamma_0 + \rho$. This further reduces the number of inner iterations in our experiments.

We briefly list the problem classes used for our experiments. Detailed descriptions can be found in the appendix. "OA" and "OD" correspond to optimal design of experiment problems with A- and D-criterion respectively. "Port" are portfolio instances with a log-revenue objective. "QuadProb" and "Ill" are quadratic instances over the simplex with a low and high condition number respectively. "Birkhoff", "Spec" and "Nuclear" are quadratic instances over the Birkhoff polytope, the spectraplex, and a nuclear norm ball respectively.

We benchmark the performance of SLS against the adaptive step-size strategies of Pedregosa et al. (2020); Pokutta (2024), the golden-ratio and backtracking line search strategies, the monotonic step size of Carderera et al. (2021; 2024), and the agnostic step size, using the implementations from the `FrankWolfe.jl` package. All strategies are run with the target accuracy of $10^{-7}$, compatible with the gap stopping criterion we set. In the main part, we compare SLS against the adaptive step size (default strategy in `FrankWolfe.jl`), the agnostic step size, and the backtracking line search. All other experiments with much more detailed reporting are available in Section C.

Fig. 2 illustrates the evolution of the step size from the adaptive and secant strategies on four instances on which the secant method converged to the desired FW gap. The convergence of SLS is notably faster on the nuclear norm and portfolio instances, while maintaining higher step sizes during the first iterations. On the D-OED and quadratic problems, we also notice that the step size computed by secant oscillates less than the adaptive line search.

Fig. 1 shows the number of iterations needed by SLS to converge on all problem classes, the total number of instances per problem class can be found in Table 1. For most quadratic problems (Birkhoff, Ill, QuadProb, Spec), SLS requires at most one iteration in accordance with Remark 3.4. Quadratics on the nuclear norm ball are the exception, with some instances taking more than one secant iteration per line search on average. This can be traced back to numerical issues, since on most instances the primal gap is quickly below $10^{-10}$ but the dual gap remains large due to extremely large nuclear norm ball radii. Importantly, the average iteration count remains low, around 1.5, for all solved instances

even in non-quadratic cases. High average iteration counts correspond to instances causing numerical instabilities.

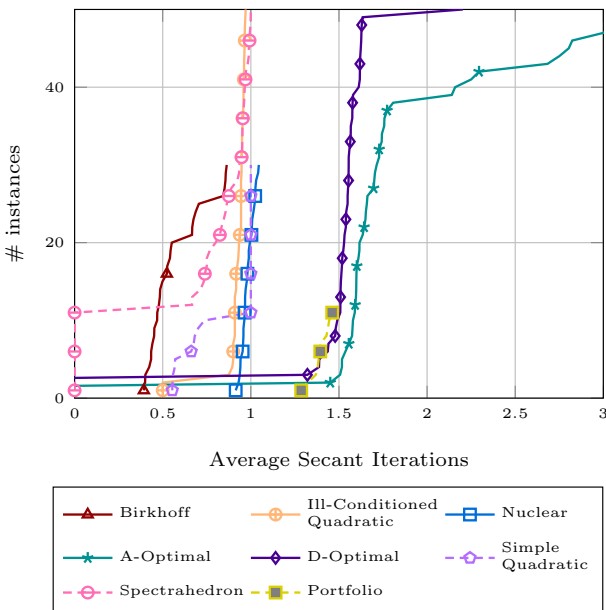

*Figure 1.* Average secant iteration count by problem class. The graph is truncated at 3 iterations on the right, only affecting the $A$-Optimal instances, which are known to be numerically unstable (Hendrych et al., 2024) with very flat descent directions. Note that the average number can be lower than 1 due to warm-starting.

Aggregated results for all instance classes are compiled in Table 1 and broken down in more detail in the appendix. Note that the agnostic step size does not ensure convergence on non-smooth functions such as OA and OD. The SLS strategy outperforms the two other methods in number of FW iterations, gap and time on most instance classes. It is particularly interesting that SLS outperforms an essentially free to compute step size such as the agnostic on instance classes on which fast convergence rates can be guaranteed, such as a squared distance objective on a polytope. While SLS is not systematically the fastest strategy for all instance types, it is often very close to the best option, making it an effective and robust choice for a wide spectrum of problems.

Detailed trajectories of the most important step-size strategies are illustrated in Fig. 3 for four instances of different types. They show the superior performance of SLS compared to backtracking line search, to the first-order adaptive strategy of Pokutta (2024), and open-loop step sizes. We compare here against the open-loop strategy because of its popularity for Frank-Wolfe, even though it is not known to converge on non-smooth self-concordant functions such as the OD and portfolio objectives. We highlight that the performance against the number of iterations carries over to the convergence against wall-clock time, showing that SLS is performant to minimize the function on the segment,

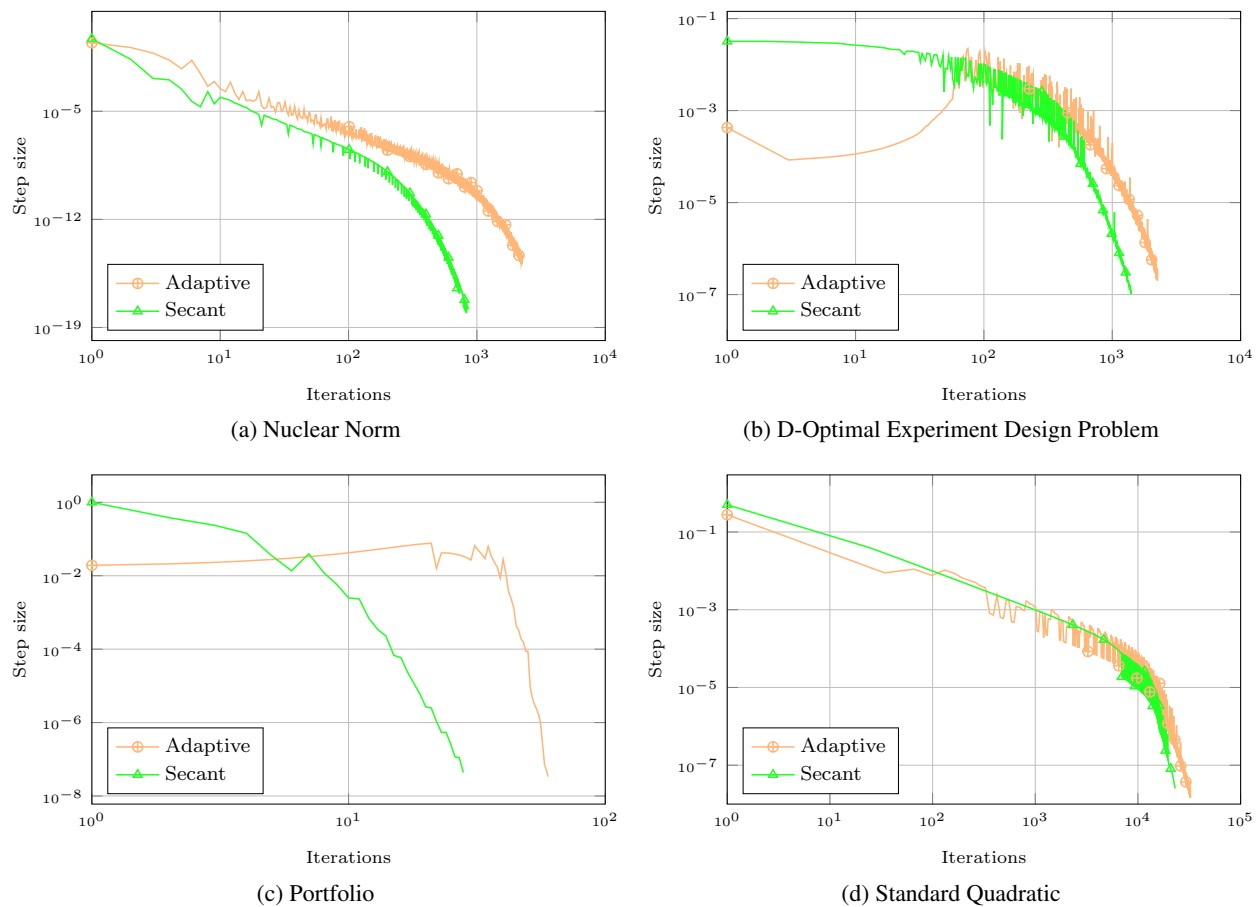

*Figure 2.* Comparison of the step sizes per iteration for the Secant and Adaptive line search.

but also efficient in terms of computations despite requiring gradient calls, unlike, e.g., backtracking which evaluates the function only. On the numerically-challenging nuclear norm ball example in particular, backtracking quickly stagnates despite theoretically being an equivalent line search.

*Table 1.* Summary of the performance on all considered problems. The number of instances per problem classes are written in the brackets. The geometric mean of the solving time is taken over all instances. The geometric mean of the dual gap is only taken over instances that could not be solved up to the tolerance. The average number of iterations is taken over all solved instances.

| | Secant | | | Adaptive | | | Agnostic | | | Backtracking | | |
|---|---|---|---|---|---|---|---|---|---|---|---|---|
| Prob. | Time (s) | Dual gap | # FW iterations | Time (s) | Dual gap | # FW iterations | Time (s) | Dual gap | # FW iterations | Time (s) | Dual gap | # FW iterations |
| Birk. [30] | 427.6 | <1e-7 | 10624 | 614.6 | 2.47e-7 | 17506 | 2959.9 | 8.76e-7 | 731017 | **248.5** | <1e-7 | **7283** |
| Ill [50] | 15.3 | <1e-7 | **20** | 15.9 | <1e-7 | 26 | 1227.7 | 4.35e-6 | 120361 | **13.9** | <1e-7 | 34 |
| Nuclear [30] | **204.6** | **0.133** | **1071** | 600.6 | 0.139 | 2547 | 3600.0 | 753000.0 | – | 3600.0 | 11400.0 | – |
| OA [50] | **475.7** | 0.0333 | **2194** | 3600.0 | 0.0935 | – | 3600.0 | 0.13 | – | 3600.0 | **5.02e-5** | – |
| OD [50] | 199.2 | 8.08e-5 | 828 | 252.0 | 2.90e-7 | 1496 | 3542.2 | 3.83e-4 | >9M | **158.8** | **<1e-7** | **748** |
| Port [13] | **0.5** | **<1e-7** | **28** | 4.6 | 1.44e-7 | 66 | 2083.7 | 2.09e-06 | 1263 | 166.8 | 3.63e-7 | 40 |
| QuadProb [30] | 340.7 | 0.000217 | **15046** | 355.7 | 0.000237 | 20814 | 3043.1 | **0.000163** | >30M | 426.4 | 0.000624 | 32585 |
| Spec [50] | **34.5** | **1.77e-7** | **124** | 41.1 | 3.8e-7 | 139 | 192.3 | 2.13e-6 | 4252 | 168.7 | 1.35e-6 | 1347 |

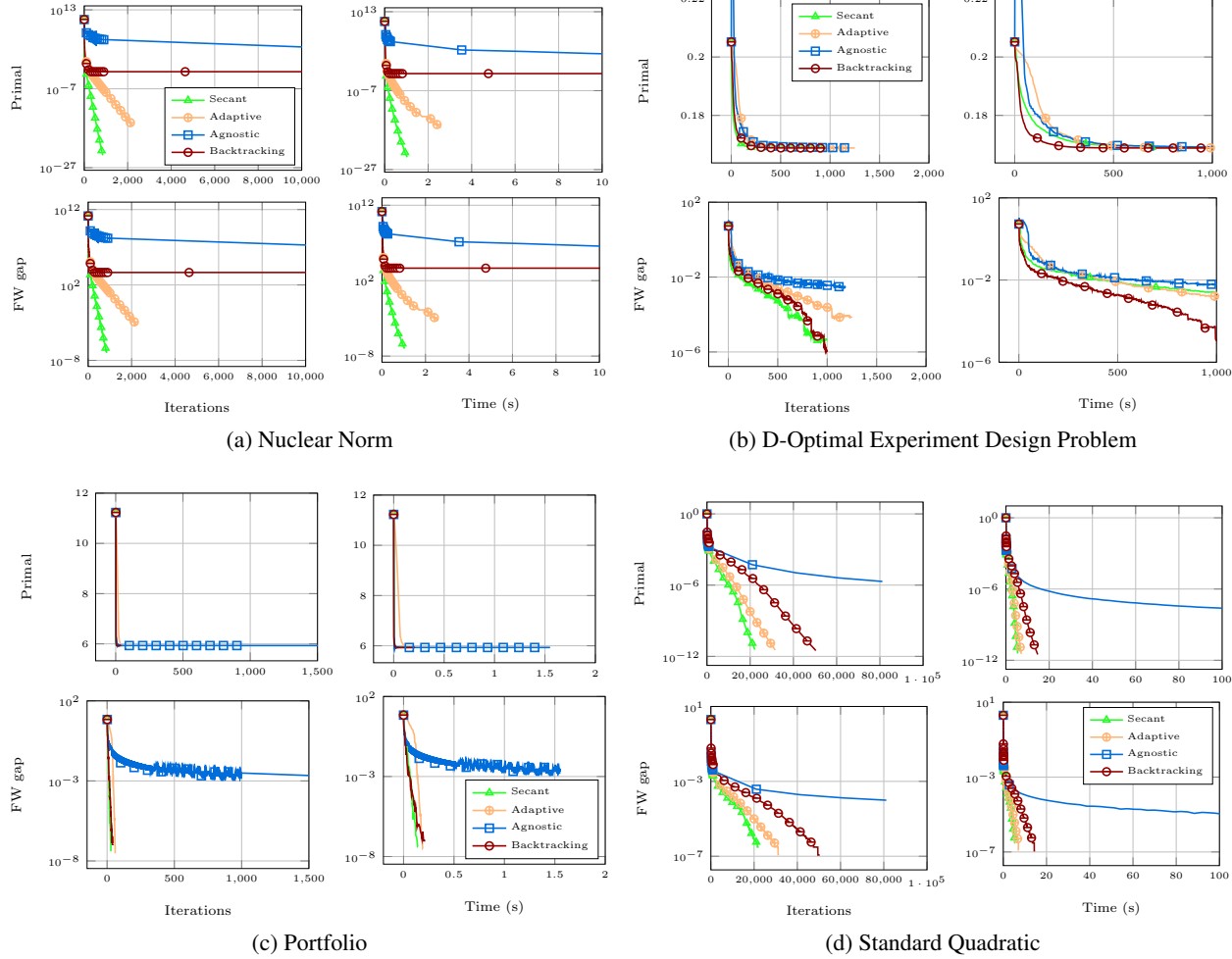

(a) Nuclear Norm

(b) D-Optimal Experiment Design Problem

(c) Portfolio

(d) Standard Quadratic

*Figure 3.* Progress of the primal value and FW gap for two problems with a quadratic objective (nuclear norm and standard quadratic) and for two problems with a self-concordant objective (D-Opt and portfolio). In (a), the iterations and time are truncated at 10000 and 10 s, repectively, since Backtracking and Agnostic stall on this instance.

## Impact Statement

This work introduces a novel step-size strategy for Frank-Wolfe algorithms, leveraging the secant method. By reformulating the line search problem as a root-finding task, our approach significantly reduces computational costs while maintaining effectiveness and adaptivity to local smoothness properties. This innovation not only enhances the performance of Frank-Wolfe algorithms but also provides a framework that could be extended to other optimization methods, potentially transforming fields that rely on constrained optimization, such as operations research, machine learning, and engineering design. The broader implications of this work lie in its ability to bridge theoretical and practical optimization challenges. By offering a computationally efficient and theoretically sound step-size strategy, we enable researchers and practitioners to tackle larger and more complex problems than previously feasible. This could lead to advancements in areas such as resource allocation, network design, and large-scale machine learning, where constrained optimization is ubiquitous. Ethically, this work aligns with the goal of advancing open scientific inquiry and computational efficiency, and we do not foresee any ethical issues; the same holds true for immediate social impact.

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

# A. Precision of FW's Line Search

In this section, we analyze the convergence of the FW algorithm when there is an inexact computation of the line search, depending on the choice of the accuracy parameters.

Assume $f : \mathbb{R}^n \to \mathbb{R}$ is convex, and differentiable in an open set containing the convex and compact feasible set $\mathcal{X}$, and let $D \stackrel{\text{def}}{=} \operatorname{diam}(\mathcal{X})$. Also, assume $f$ is $L$-smooth in $\mathcal{X}$, that is,

$$f(\mathbf{y}) \le f(\mathbf{x}) + \langle \nabla f(\mathbf{y}), \mathbf{x} - \mathbf{y} \rangle + \frac{L}{2} \|\mathbf{x} - \mathbf{y}\|^2, \text{ for all } \mathbf{x}, \mathbf{y} \in \mathcal{X}.$$

---

**Algorithm 3** Frank-Wolfe with inexact line search

---

**Input:** Function $f$, initial point $\mathbf{x}_0$.
1: **for** $t = 0$ **to** $T$ **do**
2: $\quad \mathbf{v}_t \leftarrow \underset{\mathbf{v} \in \mathbb{R}^n}{\operatorname{argmin}} \langle \nabla f(\mathbf{x}_t), \mathbf{v} \rangle$
3: $\quad \mathbf{x}_{t+1} \leftarrow \delta_t$-minimizer of $g : [0,1] \to \mathbb{R}, \gamma \mapsto f((1-\gamma)\mathbf{x}_t + \gamma \mathbf{v}_t)$
4: **end for**

---

We define positive weights $a_\ell > 0$ for all $\ell \ge 0$, to be determined later, and $A_t \stackrel{\text{def}}{=} \sum_{i=0}^{t} a_i$. Thus, $A_{-1} = 0$. We also define the following lower bound on $f(\mathbf{x}^*)$, where $\mathbf{x}^*$ here is defined as a minimizer in $\operatorname{argmin}_{\mathbf{x} \in \mathcal{X}} f(\mathbf{x})$:

$$A_t f(\mathbf{x}^*) \overset{\text{\textcircled{1}}}{\ge} \sum_{\ell=0}^{t} a_\ell f(\mathbf{x}_\ell) + \sum_{\ell=0}^{t} a_\ell \langle \nabla f(\mathbf{x}_\ell), \mathbf{x}^* - \mathbf{x}_\ell \rangle$$

$$\overset{\text{\textcircled{2}}}{\ge} \sum_{\ell=0}^{t} a_\ell f(\mathbf{x}_\ell) + \sum_{\ell=0}^{t} a_\ell \langle \nabla f(\mathbf{x}_\ell), \mathbf{v}_\ell - \mathbf{x}_\ell \rangle$$

$$\stackrel{\text{def}}{=} A_t L_t,$$

where we applied convexity in ①, and for ②, we applied the definition of $\mathbf{v}_\ell$ in the algorithm, which is why we define this point. Define the primal-dual gap

$$G_t \stackrel{\text{def}}{=} f(\mathbf{x}_{t+1}) - L_t \ge f(\mathbf{x}_{t+1}) - f(\mathbf{x}^*), \tag{A.1}$$

and note that upper bound $f(\mathbf{x}_{t+1})$ is one step ahead, which helps the analysis. We obtain the following, for $t \ge 0$, and $\tilde{\mathbf{x}}_{t+1} \stackrel{\text{def}}{=} \frac{A_{t-1}}{A_t} \mathbf{x}_t + \frac{a_t}{A_t} \mathbf{v}_t$:

$$A_t G_t - A_{t-1} G_{t-1} = A_t f(\mathbf{x}_{t+1}) - A_{t-1} f(\mathbf{x}_t)$$

$$- \left( a_t f(\mathbf{x}_t) + \sum_{\ell=0}^{t-1} a_\ell f(\mathbf{x}_\ell) + a_t \langle \nabla f(\mathbf{x}_t), \mathbf{v}_t - \mathbf{x}_t \rangle + \sum_{\ell=0}^{t-1} a_\ell (\langle \nabla f(\mathbf{x}_\ell), \mathbf{v}_\ell - \mathbf{x}_\ell \rangle) \right)$$

$$+ \left( \sum_{\ell=0}^{t-1} a_\ell f(\mathbf{x}_\ell) + \sum_{\ell=0}^{t-1} a_\ell (\langle \nabla f(\mathbf{x}_\ell), \mathbf{v}_\ell - \mathbf{x}_\ell \rangle) \right) \tag{A.2}$$

$$\overset{\text{\textcircled{1}}}{\le} \langle \nabla f(\mathbf{x}_t), A_t(\tilde{\mathbf{x}}_{t+1} - \mathbf{x}_t) - a_t(\mathbf{v}_t - \mathbf{x}_t) \rangle + \frac{LA_t}{2} \|\tilde{\mathbf{x}}_{t+1} - \mathbf{x}_t\|^2 + A_t \delta_t$$

$$\overset{\text{\textcircled{2}}}{=} \frac{La_t^2}{2A_t} \|\mathbf{v}_t - \mathbf{x}_t\|^2 + A_t \delta_t \overset{\text{\textcircled{3}}}{\le} \frac{LD^2 a_t^2}{2A_t} + A_t \delta_t \stackrel{\text{def}}{=} E_t.$$

In ① we grouped terms to get $A_t(f(\mathbf{x}_{t+1}) - f(\mathbf{x}_t))$ used that the assumption on $\mathbf{x}_{t+1}$ makes this term be bounded by the value that the smoothness inequality yields for any point, and in particular for $\tilde{\mathbf{x}}_{t+1}$, up to a $\delta_t$ error coming from the line search. In ② we used twice the definition of $\tilde{\mathbf{x}}_{t+1}$ which implies $A_t(\mathbf{x}_{t+1} - \mathbf{x}_t) = a_t(\mathbf{v}_t - \mathbf{x}_t)$. In ③, we bounded the distance of points by the diameter $D$.

Now with the choice $a_t = 2t + 2$ and $A_t = \sum_{i=0}^{t} a_i = (t+1)(t+2)$ we have, by telescoping and dividing by $A_t$:

$$f(\mathbf{x}_{t+1}) - f(\mathbf{x}^*) \leq G_t \leq \frac{1}{A_t} \sum_{i=0}^{t} E_i = \frac{1}{A_t} \sum_{i=0}^{t} \frac{LD^2 a_i^2}{2A_i} + A_i \delta_i = \frac{1}{(t+1)(t+2)} \sum_{i=0}^{t} \frac{2LD^2(i+1)}{i+2} + \frac{1}{A_t} \sum_{i=0}^{t} A_i \delta_i$$

$$< \frac{2LD^2}{t+2} + \frac{1}{A_t} \sum_{i=0}^{t} A_i \delta_i.$$

(A.3)

If we choose constant $\delta_t = \delta$, the sum of the terms with $\delta$ on the right hand side becomes $O(t\delta)$. So if we are aiming to get an $\epsilon$-minimizer after $T$ steps, we can choose $T$ so that the first summand on the right hand side is $\epsilon/2$ and we can choose $\delta = \Theta(\epsilon/T)$ so that the second summand is $\epsilon/2$ after $T$ steps.

Another option would be to choose $\delta_i = \frac{\epsilon a_i}{2A_i}$, which yields $\frac{1}{A_t} \sum_{i=0}^{t} A_i \delta_i = \frac{\epsilon}{2A_t} \sum_{i=0}^{t} a_i = \frac{\epsilon}{2}$ after any number of steps.

## B. Experimental Results for Secant Method vs Newton's Method

A natural question to ask is whether we can replace the secant method with the Newton's method in SLS for higher performance, provided that we have access to second-order information. It turns out that apart from requiring additional second-order ormation that may not be available, even if available and computed via automatic differentiation or provided directly, the secant method is faster than the Newton's method in practice in our application context.

To this end we ran the following computational experiments. We run SLS with secant method and Newton's method in isolation outside of the Frank-Wolfe algorithm to avoid (1) any overheads that may arise and (2) side effects due to different trajectories that the algorithms may take and that propagate, so that different line search problems would be solved.

For this experiment, we performed two different types of experiments. We first ran both methods on a subset of the instances from Badr et al. (2022), as they cover a broad range of cases, also outside our direct application context. We ran each variant 1000 times for each instance and report total timings. Both methods were run with the same initial step size $\gamma_0 = 0$ and $\gamma_1 = \gamma_0 + \rho$ with $\rho = 10^{-5}$ and to the target accuracy of $10^{-8}$. For these tests, we computed the gradient function via automatic differentiation. We report the average time in seconds for each method to find a solution with a relative accuracy of $10^{-5}$.

We then ran the two methods on 1000 randomly generated line-search instances for two relevant cases that typically occur in the context of the Frank-Wolfe algorithm. For the case where $f$ is a quadratic function, we used the same instance generation as done in Section 4 for the quadratics and randomly sampled $\mathbf{x}$ and $\mathbf{d}$ from the unit sphere. For the case where $f$ is a non-quadratic function, we derive the line search problems from the portfolio optimization instances. In the quadratic case, the gradient has been provided directly and in the case of the non-quadratic functions, we used finite differences to compute the gradient; to more closely mimic the line search problems that we encounter.

We report the timings in Table 2 and provide some visualizations of the considered problems in Fig. 4.

## C. Additional Experiments

In this appendix, we provide additional information and context on the instance classes, and report more fine-grained results on experiments by problem class.

**Optimal Design of Experiment (OED).** Optimal Design of Experiment is a problem maximizing an information criterion on the probability simplex and which was tackled by a FW method in Hendrych et al. (2024). The objective functions of both the 'A' and 'D' criteria are generalized self-concordant, not globally smooth, and result in two groups of instances which we denote by "OA" and "OD" respectively.

**Portfolio Optimization.** We consider a portfolio optimization model with log-revenue similar to Dvurechensky et al. (2023); Carderera et al. (2021; 2024). The instances are taken from Carderera et al. (2024) and denoted by "Port". The objective function is also (generalized) self-concordant and satisfies the conditions of Remark 3.2.

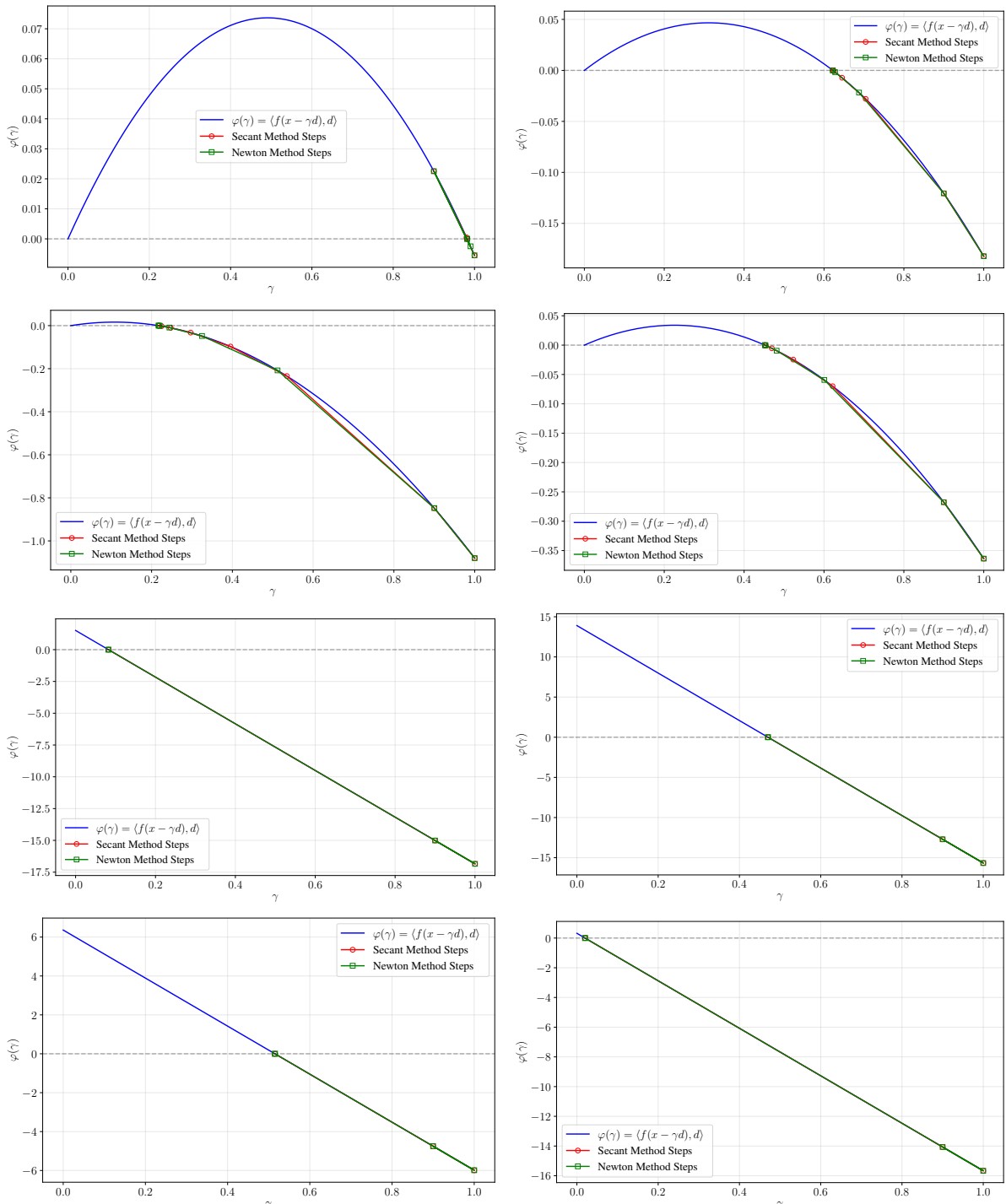

*Figure 4.* Various runs of SLS with the secant and Newton's method. (Top Row) Instances where $f$ is not a quadratic function. (Bottom Row) Instances where $f$ is a quadratic function.

*Table 2.* Comparison of execution times between Secant and Newton's methods for various test functions. Where $f$ is explicitly provided, the gradient function is computed via automatic differentiation. For the line search problems, the gradient is provided directly and not computed via automatic differentiation, as those are instances extracted from the line search routine. Timings are averaged over 1000 runs.

| Function | Secant Time (s) | Newton's Time (s) | Newton/Secant |
|---|---|---|---|
| $f(x) = x^2 - 2$ | 3.42e-06 | 4.32e-05 | 12.62 |
| $f(x) = x^2 - 5$ | 3.14e-06 | 5.47e-05 | 17.42 |
| $f(x) = x^2 - 10$ | 2.82e-06 | 4.13e-05 | 14.65 |
| $f(x) = x^2 - x - 2$ | 3.70e-06 | 4.63e-05 | 12.52 |
| $f(x) = x^2 + 2x - 7$ | 3.47e-06 | 4.11e-05 | 11.85 |
| $f(x) = x^3 - 2$ | 4.81e-06 | 4.64e-05 | 9.65 |
| $f(x) = xe^x - 7$ | 1.01e-05 | 5.89e-05 | 5.83 |
| $f(x) = x - \cos(x)$ | 6.39e-06 | 4.39e-05 | 6.88 |
| Line Search (quadratic) | 1.92e-05 | 7.97e-05 | 4.15 |
| Line Search (arbitrary) | 3.75e-05 | 1.24e-04 | 3.31 |

**Quadratics on the simplex.** Ill-conditioned problems are constructed as convex quadratic minimization on the simplex with a high condition number on the Hessian and denoted "Ill". Well-conditioned quadratic problems denoted "QuadProb" are built using the squared Euclidean distance to a random point over the simplex.

**Optimization on the spectraplex and nuclear norm ball.** In order to study the behavior of SLS on non-polyhedral constraint sets, we apply it to matrix completion problems in the symmetric and nonsymmetric cases, i.e., optimizing over the spectraplex and nuclear norm ball and respectively denoted with "Spec" and "Nuclear".

**Birkhoff polytope.** The problem class "Birkhoff" corresponds to minimizing a least-square objective on the Birkhoff polytope, the convex hull of all permutation matrices of a given size.

We compare more step-size strategies, including the backtracking and golden-ratio line-search strategies, the adaptive step size with zero-th order and first-order information from Pedregosa et al. (2020) and Pokutta (2024) respectively, the monotonic step size designed for generalized self-concordant functions in Carderera et al. (2021; 2024), and the open-loop agnostic step size of the form $\frac{2}{t+2}$.

Figures 5 to 12 present trajectories of the primal value and FW gaps for instances of all problem classes using the different step-size strategies. We notice on the ill-conditioned and simple quadratic problems in Fig. 6 and Fig. 11 respectively that then golden-ratio line search starts with a trajectory similar to the backtracking and secant step sizes but start to stagnate due to numerical inaccuracies earlier before reaching the desired FW gap tolerance. This is due to its stopping criterion incurring a higher additive error on the step size.

We primarily used the Blended Pairwise Conditional Gradients (BPCG) algorithm from Tsuji et al. (2022)—the default active-set algorithm in `FrankWolfe.jl`—for all experiments. Similar to the Away-step Frank-Wolfe (see e.g., Lacoste-Julien & Jaggi (2015)), it requires a step size computed with an exact or approximate line search, in particular to ensure accelerated convergence rates in favorable cases (e.g., uniformly convex functions or uniformly convex feasible regions). Note that in particular for BPCG the backtracking line search sometimes favorably interacts with the drop step mechanics of the BPCG algorithm, leading unexpected performance improvements.

For completeness, we also present experiments for SLS applied to the standard FW algorithm as implemented in `FrankWolfe.jl` in Fig. 14 and Fig. 15.

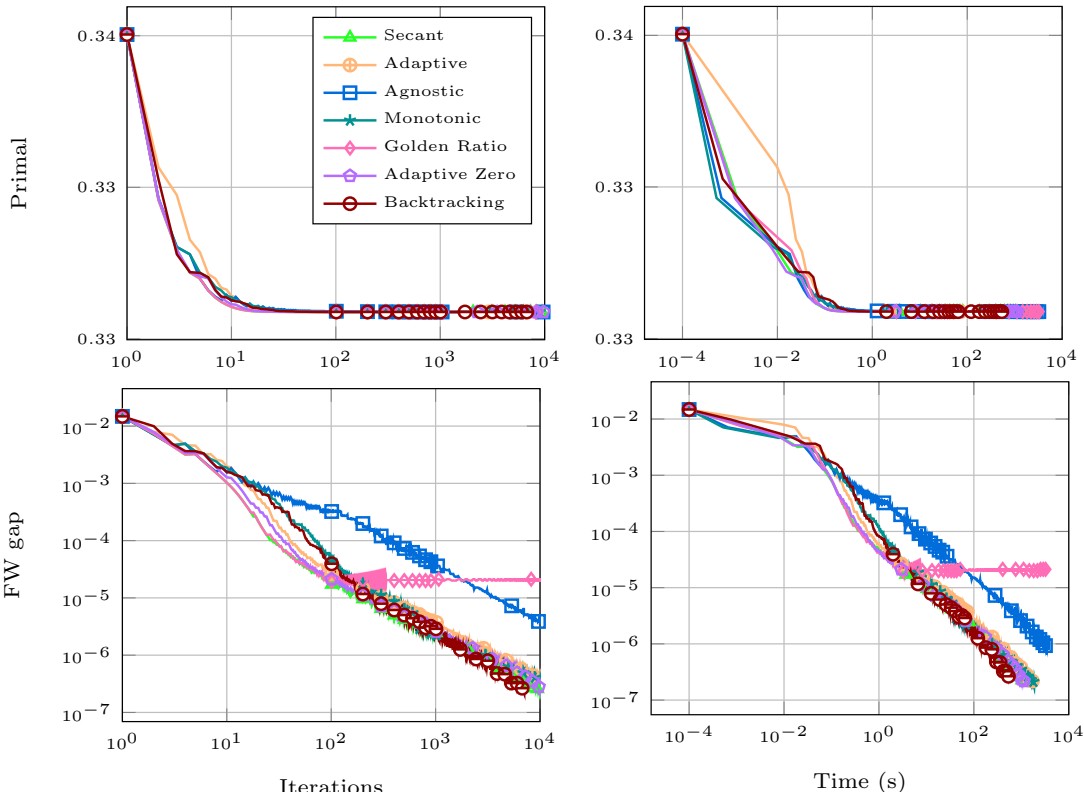

*Figure 5.* Progress of the primal value and FW gap for an instance of the Birkhoff problem. The Golden Ratio line search from the `FrankWolfe.jl` package stalls out due to numerical issues.

*Table 3.* Instances of the Birkhoff problem ordered by difficulty. The geometric mean of the solving time is taken over all instances. The geometric mean of the dual gap is only taken over instances that could not be solved up to the tolerance. The average number of iterations is taken over all solved instances.

| | | Secant | | | | Adaptive | | | | Agnostic | | | | Backtracking | | | |
|---|---|---|---|---|---|---|---|---|---|---|---|---|---|---|---|---|---|
| Dim | # | Time (s) | Dual gap | # FW iterations | Iter per sec | Time (s) | Dual gap | # FW iterations | Iter per sec | Time (s) | Dual gap | # FW iterations | Iter per sec | Time (s) | Dual gap | # FW iterations | Iter per sec |
| 2500 | 5 | 10.4 | <1e-7 | 6274 | 603.3 | 14.6 | <1e-7 | 9768 | 669.0 | 1111.7 | <1e-7 | 731017 | 657.6 | **6.0** | <1e-7 | **3919** | 653.2 |
| 10000 | 5 | 133.6 | <1e-7 | 12063 | 90.3 | 209.7 | <1e-7 | 19437 | 92.7 | 3600.0 | 2.01e-07 | – | 97.0 | **70.5** | <1e-7 | **7547** | 107.0 |
| 22500 | 5 | 523.2 | <1e-7 | 13835 | 26.4 | 818.6 | <1e-7 | 22589 | 27.6 | 3600.0 | 4.53e-07 | – | 30.1 | **284.5** | <1e-7 | **9123** | 32.1 |
| 40000 | 5 | 1137.4 | <1e-7 | 11965 | 10.5 | 1838.5 | <1e-7 | 19443 | 10.6 | 3600.0 | 8.71e-07 | – | 12.1 | **658.1** | <1e-7 | **8392** | 12.8 |
| 62500 | 5 | 2022.4 | <1e-7 | 10272 | 5.1 | 3043.7 | <1e-7 | 16293 | 5.4 | 3600.0 | 1.28e-06 | – | 6.6 | **1130.5** | <1e-7 | **6994** | 6.2 |
| 90000 | 5 | 3344.8 | <1e-7 | 9333 | 2.8 | 3600.0 | 2.47e-07 | – | 3.1 | 3600.0 | 1.58e-06 | – | 4.5 | **2253.3** | <1e-7 | **7722** | 3.4 |

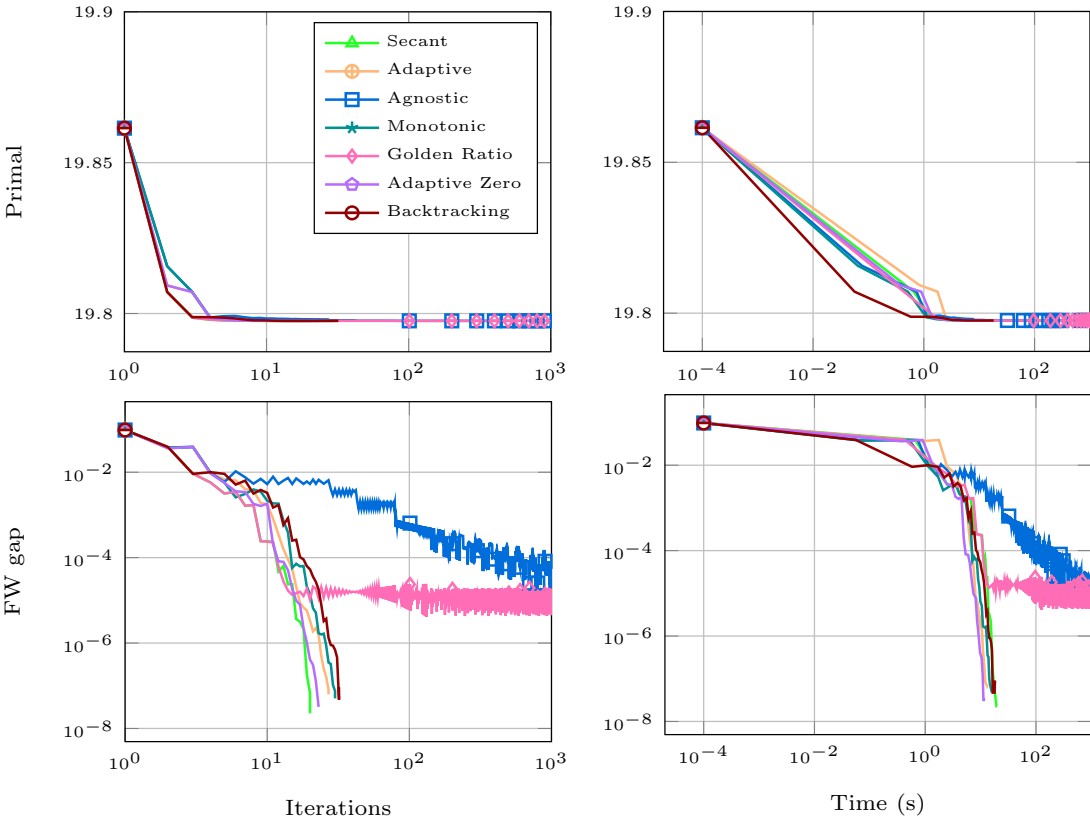

*Figure 6.* Progress of the primal value and FW gap for an instance of the Ill-Conditioned Quadratic problem. For the golden ratio line search the same remark applies as in the Birkhoff problem (see Fig. 5).

*Table 4.* Instances of the Ill-Conditioned Quadratic problem ordered by difficulty. The geometric mean of the solving time is taken over all instances. The geometric mean of the dual gap is only taken over instances that could not be solved up to the tolerance. The average number of iterations is taken over all solved instances.

| | | Secant | | | | Adaptive | | | | Agnostic | | | | Backtracking | | | |
|---|---|---|---|---|---|---|---|---|---|---|---|---|---|---|---|---|---|
| Dim | # | Time (s) | Dual gap | # FW iterations | Iter per sec | Time (s) | Dual gap | # FW iterations | Iter per sec | Time (s) | Dual gap | # FW iterations | Iter per sec | Time (s) | Dual gap | # FW iterations | Iter per sec |
| 500 | 5 | **0.0** | <1e-7 | **25** | **Inf** | 0.0 | <1e-7 | 30 | Inf | 40.7 | <1e-7 | 336307 | 8263.1 | 0.1 | <1e-7 | 39 | 390.0 |
| 1000 | 5 | 9.1 | <1e-7 | **16** | 1.8 | 9.7 | <1e-7 | 22 | 2.3 | 1385.4 | 3.14e-06 | 118 | 9.2 | **7.6** | <1e-7 | 31 | 4.1 |
| 1500 | 5 | **12.5** | <1e-7 | **21** | 1.7 | 13.1 | <1e-7 | 29 | 2.2 | 3600.0 | 1.99e-06 | – | 6.3 | 17.7 | <1e-7 | 38 | 2.1 |
| 2000 | 5 | **13.7** | <1e-7 | **17** | 1.2 | 13.0 | <1e-7 | 21 | 1.6 | 860.4 | 4.20e-06 | 309 | 8.4 | 17.2 | <1e-7 | 29 | 1.7 |
| 2500 | 5 | 23.6 | <1e-7 | **20** | 0.8 | 25.4 | <1e-7 | 27 | 1.1 | 1944.4 | 5.29e-06 | 370 | 3.5 | **22.1** | <1e-7 | 37 | 1.7 |
| 3000 | 5 | 31.3 | <1e-7 | **22** | 0.7 | 32.1 | <1e-7 | 28 | 0.9 | 2133.8 | 4.85e-06 | 501 | 2.5 | **23.3** | <1e-7 | 36 | 1.5 |
| 3500 | 5 | 17.7 | <1e-7 | **13** | 0.7 | 23.0 | <1e-7 | 20 | 0.9 | 1413.0 | 5.74e-06 | 56 | 3.5 | **15.6** | <1e-7 | 24 | 1.5 |
| 4000 | 5 | 36.4 | <1e-7 | **22** | 0.6 | 32.9 | <1e-7 | 26 | 0.8 | 3600.0 | 4.91e-06 | – | 1.7 | **27.4** | <1e-7 | 38 | 1.4 |
| 4500 | 5 | 28.1 | <1e-7 | **21** | 0.7 | 32.0 | <1e-7 | 28 | 0.9 | 744.3 | 4.52e-06 | 145 | 4.8 | **24.4** | <1e-7 | 38 | 1.6 |
| 5000 | 5 | 38.6 | <1e-7 | **22** | 0.6 | 35.2 | <1e-7 | 27 | 0.8 | 2774.2 | 4.94e-06 | 1565 | 1.8 | **25.5** | <1e-7 | 34 | 1.3 |

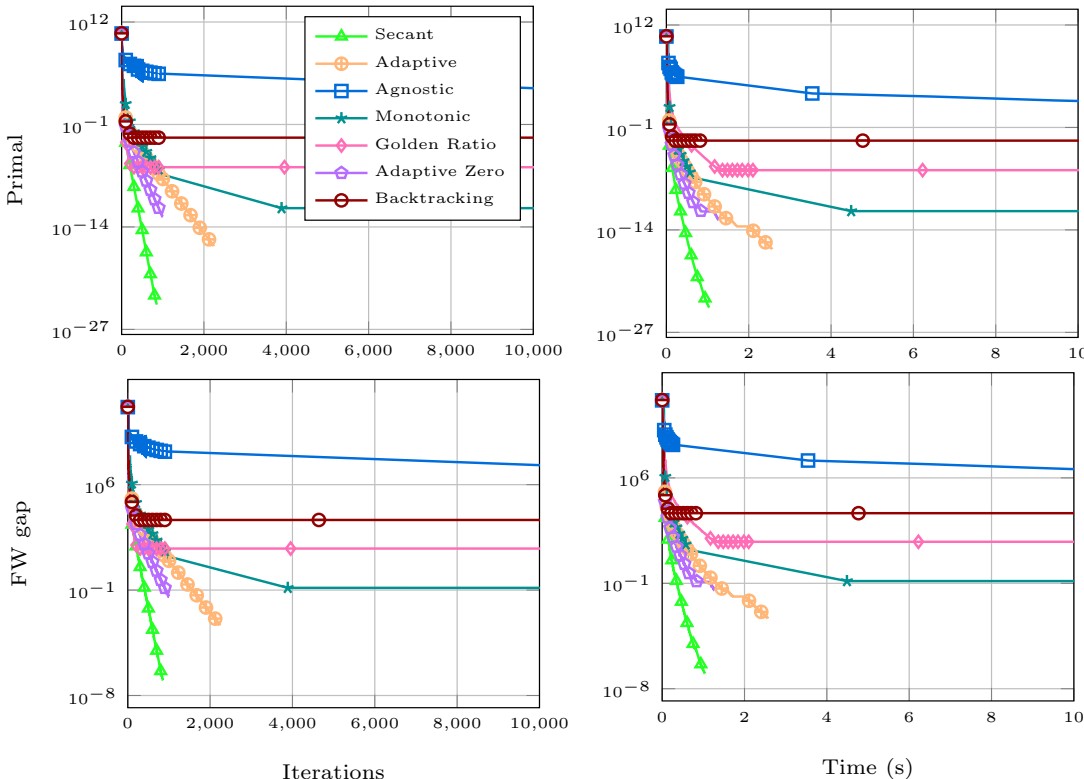

*Figure 7.* Progress of the primal value and FW gap for an instance of the Nuclear problem. The secant line search is not only significantly faster than the other methods but also reaches a much higher final accuracy.

*Table 5.* Instances of the Nuclear problem ordered by difficulty. The geometric mean of the solving time is taken over all instances. The geometric mean of the dual gap is only taken over instances that could not be solved up to the tolerance. The average number of iterations is taken over all solved instances.

| | | Secant | | | | Adaptive | | | | Agnostic | | | | Backtracking | | | |
|---|---|---|---|---|---|---|---|---|---|---|---|---|---|---|---|---|---|
| Dim | # | Time (s) | Dual gap | # FW iterations | Iter per sec | Time (s) | Dual gap | # FW iterations | Iter per sec | Time (s) | Dual gap | # FW iterations | Iter per sec | Time (s) | Dual gap | # FW iterations | Iter per sec |
| 2500 | 5 | **0.2** | <1e-7 | **511** | 2555.0 | 0.5 | <1e-7 | 1611 | 3222.0 | 3600.0 | 1.34e+04 | – | 5381.0 | 3600.0 | 1.60e+04 | – | 3316.4 |
| 10000 | 5 | **1.1** | **<1e-7** | **873** | 793.6 | 260.1 | 1.37e-01 | 3161 | 4060.2 | 3600.0 | 2.68e+04 | – | 2860.3 | 3600.0 | 2.34e+04 | – | 1250.8 |
| 22500 | 5 | **803.8** | **<1e-7** | **1662** | 2.1 | 2665.2 | 1.41e-01 | 4274 | 2.2 | 3600.0 | 3.70e+06 | – | 11.9 | 3600.0 | 1.12e+04 | – | 4.2 |
| 40000 | 5 | **2997.0** | **1.28e-01** | **1912** | 1.3 | 3600.0 | 1.38e-01 | – | 1.6 | 3600.0 | 4.27e+06 | – | 12.3 | 3600.0 | 6.60e+03 | – | 2.5 |
| 62500 | 5 | 3600.0 | **1.32e-01** | – | 1.1 | 3600.0 | 1.39e-01 | – | 1.4 | 3600.0 | 5.63e+06 | – | 10.9 | 3600.0 | 8.94e+03 | – | 1.7 |
| 90000 | 5 | 3600.0 | **1.38e-01** | – | 1.0 | 3600.0 | 1.40e-01 | – | 1.4 | 3600.0 | 5.71e+06 | – | 12.5 | 3600.0 | 8.98e+03 | – | 1.5 |

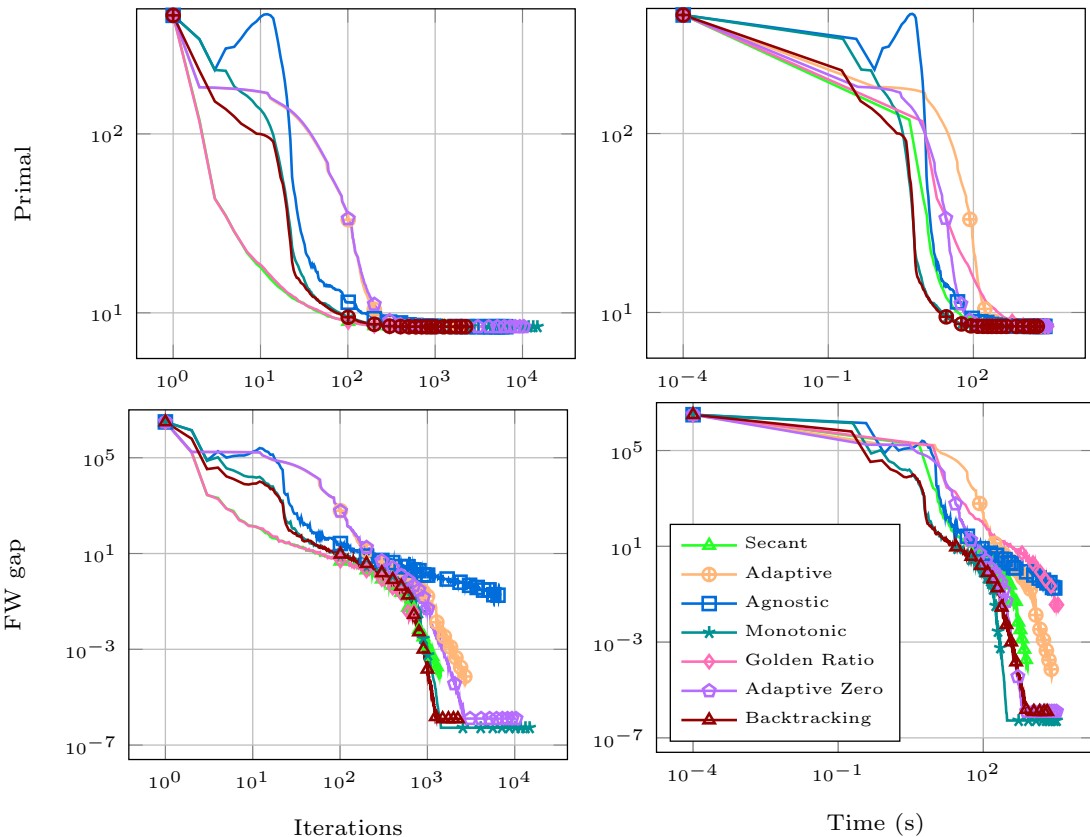

*Figure 8.* Progress of the primal value and FW gap for an instance of the A-Optimal Experiment Design problem. For the agnostic step size the same remark applies as in the D-Optimal Experiment Design problem (see Fig. 9).

*Table 6.* Instances of the A-Optimal Experiment Design problem ordered by difficulty. The geometric mean of the solving time is taken over all instances. The geometric mean of the dual gap is only taken over instances that could not be solved up to the tolerance. The average number of iterations is taken over all solved instances.

| | | Secant | | | | Adaptive | | | | Agnostic | | | | Backtracking | | | |
|---|---|---|---|---|---|---|---|---|---|---|---|---|---|---|---|---|---|
| Dim | # | Time (s) | Dual gap | # FW iterations | Iter per sec | Time (s) | Dual gap | # FW iterations | Iter per sec | Time (s) | Dual gap | # FW iterations | Iter per sec | Time (s) | Dual gap | # FW iterations | Iter per sec |
| 100 | 5 | **0.6** | **<1e-7** | **887** | 1478.3 | 3600.0 | 6.00e-07 | – | 512.2 | 3600.0 | 1.85e-05 | – | 2613.6 | 3600.0 | 1.90e-06 | – | 970.6 |
| 200 | 5 | **3.2** | **<1e-7** | **1413** | 441.6 | 3600.0 | 6.93e-07 | – | 315.1 | 3600.0 | 6.78e-05 | – | 1141.4 | 3600.0 | 1.59e-06 | – | 267.2 |
| 300 | 5 | **420.5** | **<1e-7** | **1971** | 4.7 | 3600.0 | 7.15e-07 | – | 43.9 | 3600.0 | 7.32e-03 | – | 16.2 | 3600.0 | 1.64e-06 | – | 1.9 |
| 400 | 5 | **775.3** | **<1e-7** | **2315** | 3.0 | 3600.0 | 8.87e-07 | – | 3.7 | 3600.0 | 2.16e-02 | – | 9.7 | 3600.0 | 1.82e-06 | – | 1.2 |
| 500 | 5 | **1284.7** | **<1e-7** | **2901** | 2.3 | 3600.0 | 4.47e-05 | – | 2.0 | 3600.0 | 7.51e-02 | – | 5.4 | 3600.0 | 1.56e-06 | – | 1.1 |
| 600 | 5 | **2209.3** | 2.53e-04 | **3012** | 1.1 | 3600.0 | 1.04e-06 | – | 2.7 | 3600.0 | 3.59e-02 | – | 7.7 | 3600.0 | 1.30e-06 | – | 1.0 |
| 700 | 5 | **2059.5** | **<1e-7** | **2976** | 1.4 | 3600.0 | 3.34e-03 | – | 1.5 | 3600.0 | 1.76e-01 | – | 3.2 | 3600.0 | 1.28e-06 | – | 0.9 |
| 800 | 5 | **3565.6** | 2.45e-06 | **3240** | 0.7 | 3600.0 | 2.15e-01 | – | 0.6 | 3600.0 | 2.55e-01 | – | 3.0 | 3600.0 | **1.71e-06** | – | 0.8 |
| 900 | 5 | 3600.0 | 8.13e-03 | – | 0.5 | 3600.0 | 4.53e-01 | – | 0.6 | 3600.0 | 3.50e-01 | – | 1.8 | 3600.0 | 1.41e-05 | – | 0.6 |
| 1000 | 5 | 3600.0 | 9.41e-02 | – | 0.3 | 3600.0 | 3.81e-01 | – | 0.4 | 3600.0 | 4.63e-01 | – | 1.3 | 3600.0 | **4.75e-04** | – | 0.5 |

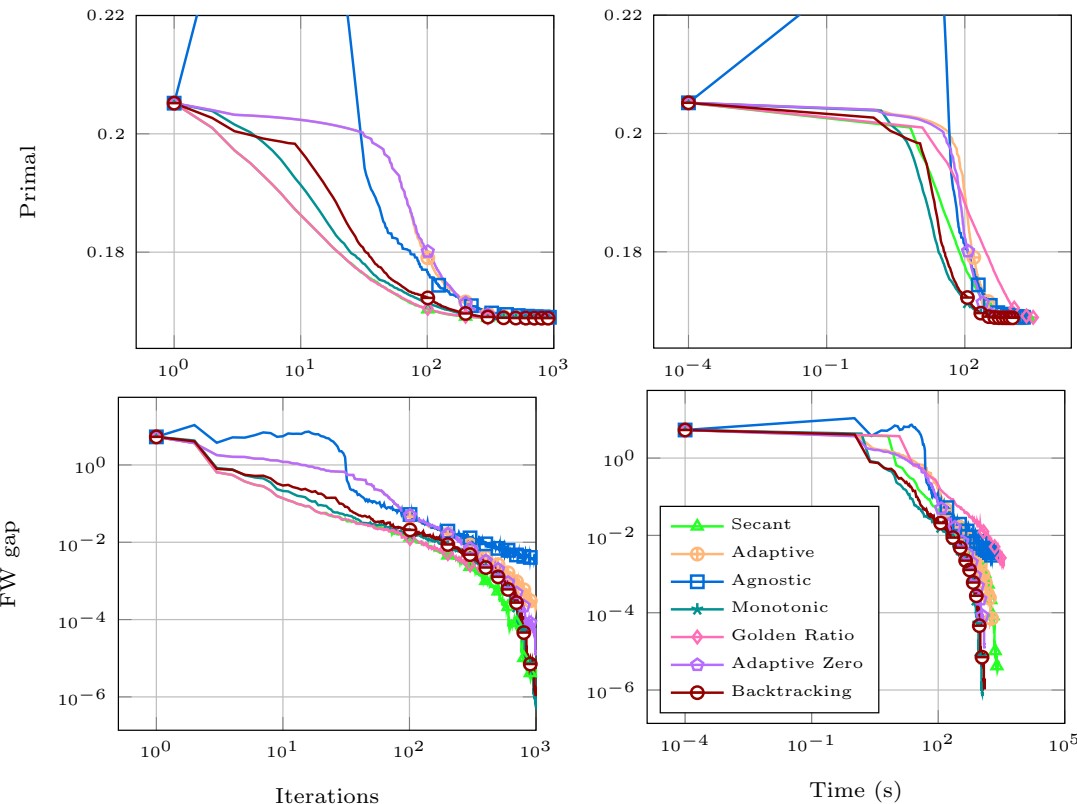

*Figure 9.* Progress of the primal value and FW gap for an instance of the D-Optimal Experiment Design problem. The significant uptick in the primal value for the agnostic is expected as (a) the agnostic step size does not guarantee primal progress in contrast to the other step sizes used here and (b) the agnostic step size strictly speaking cannot be applied to the D-Optimal Experiment Design problem (Hendrych et al., 2024) as it does not work for generalized self-concordant functions (Carderera et al., 2021; 2024). We only included it here for completeness and since it is the textbook step-size rule for the vanilla Frank-Wolfe algorithm.

*Table 7.* Instances of the D-Optimal Experiment Design problem ordered by difficulty. The geometric mean of the solving time is taken over all instances. The geometric mean of the dual gap is only taken over instances that could not be solved up to the tolerance. The average number of iterations is taken over all solved instances. Brackets indicate that all instances were basically solved up to the tolerance up to some numerical error.

| | | Secant | | | | Adaptive | | | | Agnostic | | | | Backtracking | | | |
|---|---|---|---|---|---|---|---|---|---|---|---|---|---|---|---|---|---|
| Dim | # | Time (s) | Dual gap | # FW iterations | Iter per sec | Time (s) | Dual gap | # FW iterations | Iter per sec | Time (s) | Dual gap | # FW iterations | Iter per sec | Time (s) | Dual gap | # FW iterations | Iter per sec |
| 100 | 5 | 0.2 | <1e-7 | 306 | 1530.0 | 0.3 | <1e-7 | 653 | 2176.7 | 3061.6 | [<1e-7] | >9M | 2523.4 | **0.1** | <1e-7 | **219** | 2190.0 |
| 200 | 5 | 1.1 | <1e-7 | 459 | 417.3 | 1.4 | <1e-7 | 915 | 653.6 | 3600.0 | 2.34e-07 | – | 948.8 | **0.8** | <1e-7 | **376** | 470.0 |
| 300 | 5 | 128.0 | <1e-7 | 534 | 4.2 | 64.9 | [<1e-7] | 1088 | 13.4 | 3600.0 | 1.80e-05 | – | 15.4 | **105.3** | <1e-7 | **451** | 4.3 |
| 400 | 5 | 439.7 | [<1e-7] | 780 | 1.4 | 327.2 | <1e-7 | 1423 | 4.3 | 3600.0 | 4.68e-05 | – | 9.0 | **241.3** | <1e-7 | **648** | 2.7 |
| 500 | 5 | **376.1** | <1e-7 | 812 | 2.2 | 775.4 | <1e-7 | 1570 | 2.0 | 3600.0 | 7.57e-05 | – | 6.8 | 384.8 | <1e-7 | **727** | 1.9 |
| 600 | 5 | **371.0** | <1e-7 | 876 | 2.4 | 1354.2 | [<1e-7] | 1585 | 0.9 | 3600.0 | 8.03e-05 | – | 6.9 | 415.2 | <1e-7 | **815** | 2.0 |
| 700 | 5 | 728.5 | <1e-7 | 1029 | 1.4 | 1523.1 | [<1e-7] | 1866 | 0.7 | 3600.0 | 4.74e-04 | – | 3.1 | **589.7** | <1e-7 | **912** | 1.5 |
| 800 | 5 | 1066.7 | <1e-7 | 1096 | 1.0 | 1268.0 | <1e-7 | 1986 | 1.6 | 3600.0 | 6.06e-04 | – | 2.5 | **884.8** | <1e-7 | **998** | 1.1 |
| 900 | 5 | 1881.2 | [<1e-7] | 1228 | 0.5 | 2282.9 | <1e-7 | 2177 | 1.0 | 3600.0 | 9.85e-04 | – | 1.3 | **1294.8** | <1e-7 | **1131** | 0.9 |
| 1000 | 5 | 3413.0 | 8.08e-05 | 1334 | 0.3 | 3389.9 | 2.90e-07 | 2212 | 0.7 | 3600.0 | 1.09e-03 | – | 1.0 | **1978.3** | **<1e-7** | **1199** | 0.6 |

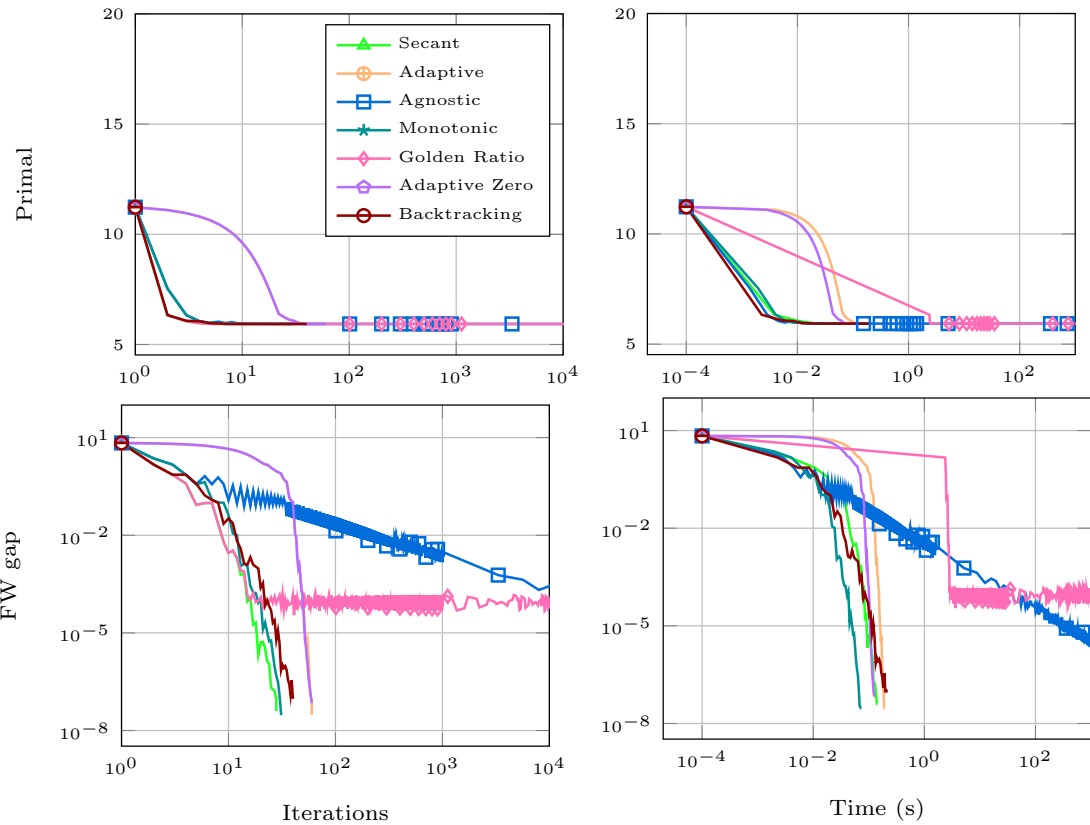

*Figure 10.* Progress of the primal value and FW gap for an instance of the Portfolio problem.

*Table 8.* Instances of the Portfolio problem ordered by difficulty. The geometric mean of the solving time is taken over all instances. The geometric mean of the dual gap is only taken over instances that could not be solved up to the tolerance. The average number of iterations is taken over all solved instances.

| | | Secant | | | | Adaptive | | | | Agnostic | | | | Backtracking | | | |
|---|---|---|---|---|---|---|---|---|---|---|---|---|---|---|---|---|---|
| Dim | # | Time (s) | Dual gap | # FW iterations | Iter per sec | Time (s) | Dual gap | # FW iterations | Iter per sec | Time (s) | Dual gap | # FW iterations | Iter per sec | Time (s) | Dual gap | # FW iterations | Iter per sec |
| 800 | 4 | 0.5 | <1e-7 | **26** | 52.0 | 0.6 | <1e-7 | 64 | 106.7 | 608.4 | 7.19e-07 | 1263 | 2872.9 | **0.2** | <1e-7 | 37 | 185.0 |
| 1200 | 4 | **0.5** | **<1e-7** | **23** | 46.0 | 11.1 | 1.63e-07 | 58 | 8339.3 | 3600.0 | 1.41e-06 | – | 445.7 | 3600.0 | 3.98e-07 | – | 88.9 |
| 1500 | 5 | **0.6** | **<1e-7** | **34** | 56.7 | 7.6 | 1.25e-07 | 74 | 14273.2 | 3600.0 | 3.45e-06 | – | 333.9 | 758.8 | 3.28e-07 | 53 | 193.9 |

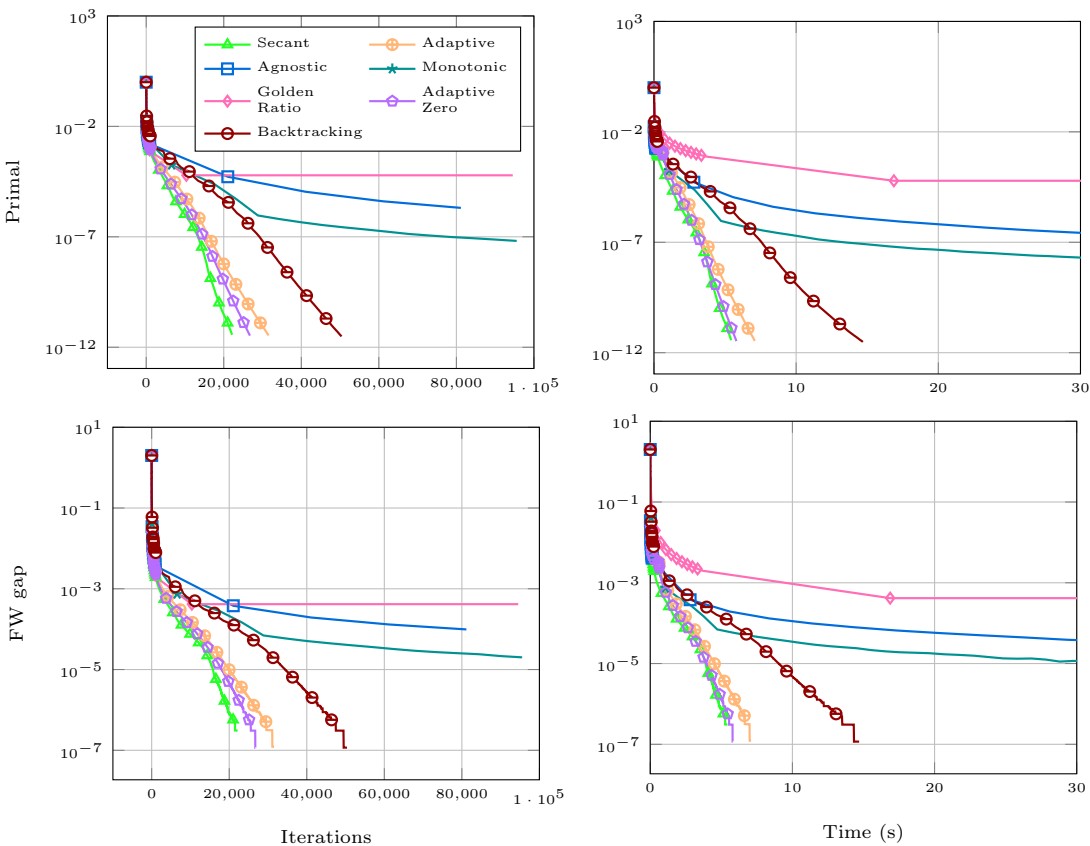

*Figure 11.* Progress of the primal value and FW gap for an instance of the Simple Quadratic problem.

*Table 9.* Instances of the Standard Quadratic problem ordered by difficulty. The geometric mean of the solving time is taken over all instances. The geometric mean of the dual gap is only taken over instances that could not be solved up to the tolerance. The average number of iterations is taken over all solved instances.

| | | Secant | | | | Adaptive | | | | Agnostic | | | | Backtracking | | | |
|---|---|---|---|---|---|---|---|---|---|---|---|---|---|---|---|---|---|
| Dim | # | Time (s) | Dual gap | # FW iterations | Iter per sec | Time (s) | Dual gap | # FW iterations | Iter per sec | Time (s) | Dual gap | # FW iterations | Iter per sec | Time (s) | Dual gap | # FW iterations | Iter per sec |
| 2500 | 5 | **0.5** | <1e-7 | **6491** | 12982.0 | 0.5 | <1e-7 | 9083 | 18166.0 | 1313.2 | <1e-7 | >30M | 23600.7 | 1.2 | <1e-7 | 13885 | 11570.8 |
| 10000 | 5 | **5.5** | <1e-7 | **23601** | 4291.1 | 6.9 | <1e-7 | 32545 | 4716.7 | 3600.0 | 3.45e-07 | – | 5558.5 | 15.6 | <1e-7 | 51285 | 3287.5 |
| 22500 | 5 | 3600.0 | **1.92e-04** | – | 2.4 | 3600.0 | 2.18e-04 | – | 3.3 | 3600.0 | 2.24e-04 | – | 10.0 | 3600.0 | 5.56e-04 | – | 3.6 |
| 40000 | 5 | 3600.0 | **2.10e-04** | – | 2.5 | 3600.0 | 2.43e-04 | – | 3.2 | 3600.0 | 2.32e-04 | – | 9.7 | 3600.0 | 5.77e-04 | – | 3.8 |
| 62500 | 5 | 3600.0 | **2.30e-04** | – | 2.4 | 3600.0 | 2.55e-04 | – | 3.2 | 3600.0 | 2.04e-04 | – | 9.1 | 3600.0 | 6.39e-04 | – | 3.6 |
| 90000 | 5 | 3600.0 | **2.36e-04** | – | 2.3 | 3600.0 | 2.34e-04 | – | 3.5 | 3600.0 | 1.56e-04 | – | 8.6 | 3600.0 | 7.26e-04 | – | 3.4 |

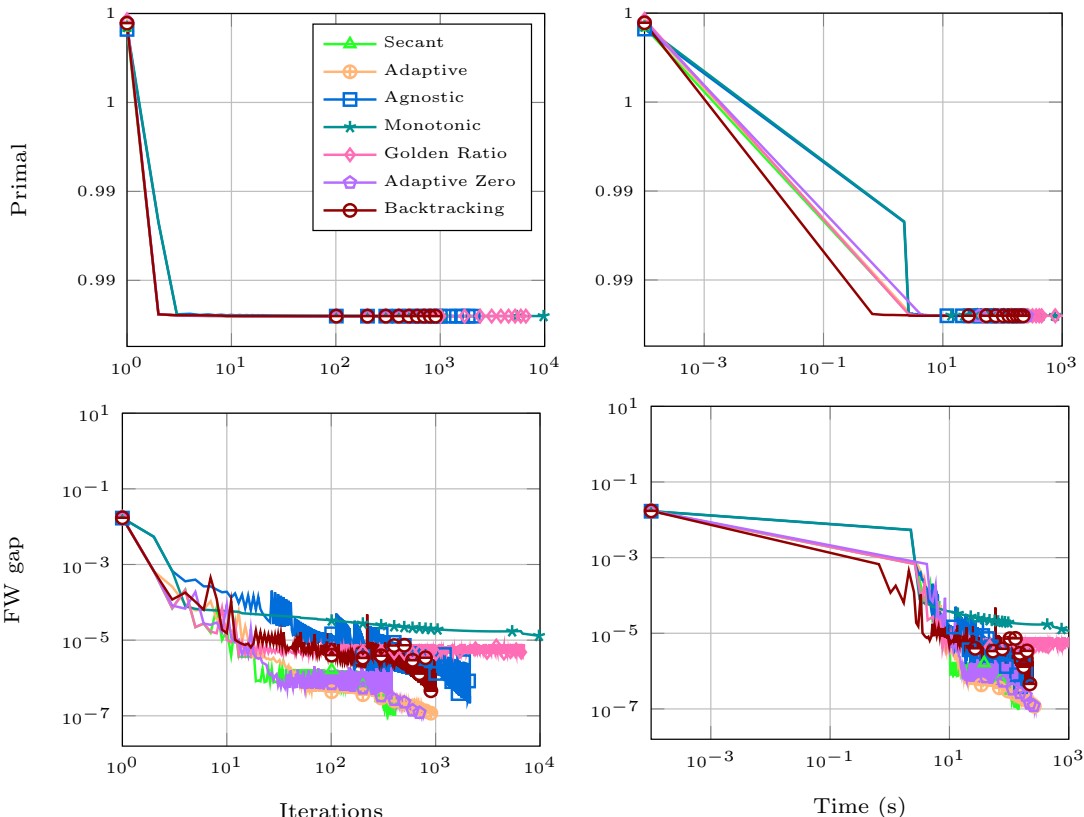

*Figure 12.* Progress of the primal value and FW gap for an instance of the Spectrahedron problem.

*Table 10.* Instances of the Spectrahedron problem ordered by difficulty. The geometric mean of the solving time is taken over all instances. The geometric mean of the dual gap is only taken over instances that could not be solved up to the tolerance. The average number of iterations is taken over all solved instances.

| | | Secant | | | | Adaptive | | | | Agnostic | | | | Backtracking | | | |
|---|---|---|---|---|---|---|---|---|---|---|---|---|---|---|---|---|---|
| Dim | # | Time (s) | Dual gap | # FW iterations | Iter per sec | Time (s) | Dual gap | # FW iterations | Iter per sec | Time (s) | Dual gap | # FW iterations | Iter per sec | Time (s) | Dual gap | # FW iterations | Iter per sec |
| 10000 | 5 | **0.4** | <1e-7 | **454** | 1135.0 | 0.4 | <1e-7 | 435 | 1087.5 | 2.5 | <1e-7 | 17323 | 6929.2 | 2.2 | <1e-7 | 4528 | 2058.2 |
| 40000 | 5 | **26.3** | <1e-7 | **400** | 15.2 | 28.2 | <1e-7 | 385 | 13.7 | 76.6 | <1e-7 | 4094 | 53.4 | 80.3 | <1e-7 | 1463 | 18.2 |
| 90000 | 5 | **15.7** | <1e-7 | **158** | 10.1 | 21.4 | <1e-7 | 300 | 14.0 | 67.4 | <1e-7 | 4442 | 65.9 | 43.1 | <1e-7 | 727 | 16.9 |
| 160000 | 5 | 11.2 | <1e-7 | 42 | 3.8 | **10.7** | <1e-7 | 46 | 4.3 | 145.2 | 3.48e-07 | 1788 | 28.1 | 100.6 | <1e-7 | 1097 | 10.9 |
| 250000 | 5 | **7.7** | <1e-7 | **6** | 0.8 | 9.3 | <1e-7 | 9 | 1.0 | 25.3 | <1e-7 | 103 | 4.1 | 41.5 | <1e-7 | 612 | 14.7 |
| 360000 | 5 | **17.4** | <1e-7 | **22** | 1.3 | 18.8 | <1e-7 | 25 | 1.3 | 523.9 | 1.73e-07 | 2462 | 9.1 | 366.6 | <1e-7 | 1472 | 4.0 |
| 490000 | 5 | 312.8 | 1.77e-07 | **10** | 0.5 | 357.5 | 4.86e-07 | 13 | 0.5 | 616.9 | 2.05e-06 | 5 | 0.4 | 577.1 | 1.10e-06 | 3 | 0.4 |
| 640000 | 5 | 78.7 | **<1e-7** | **9** | 0.1 | 100.2 | <1e-7 | 11 | 0.1 | 594.1 | 5.71e-07 | 35 | 0.3 | 678.5 | 1.43e-06 | 112 | 0.3 |
| 810000 | 5 | 525.3 | **<1e-7** | **54** | 0.1 | 587.4 | <1e-7 | 66 | 0.1 | 3600.0 | 3.83e-06 | – | 0.1 | 3600.0 | 1.65e-06 | – | 0.1 |
| 1000000 | 5 | **195.7** | **<1e-7** | **43** | 0.2 | 373.1 | 1.66e-07 | 22 | 0.2 | 1464.3 | 2.31e-06 | 47 | 0.2 | 775.2 | 1.04e-06 | 6 | 0.2 |

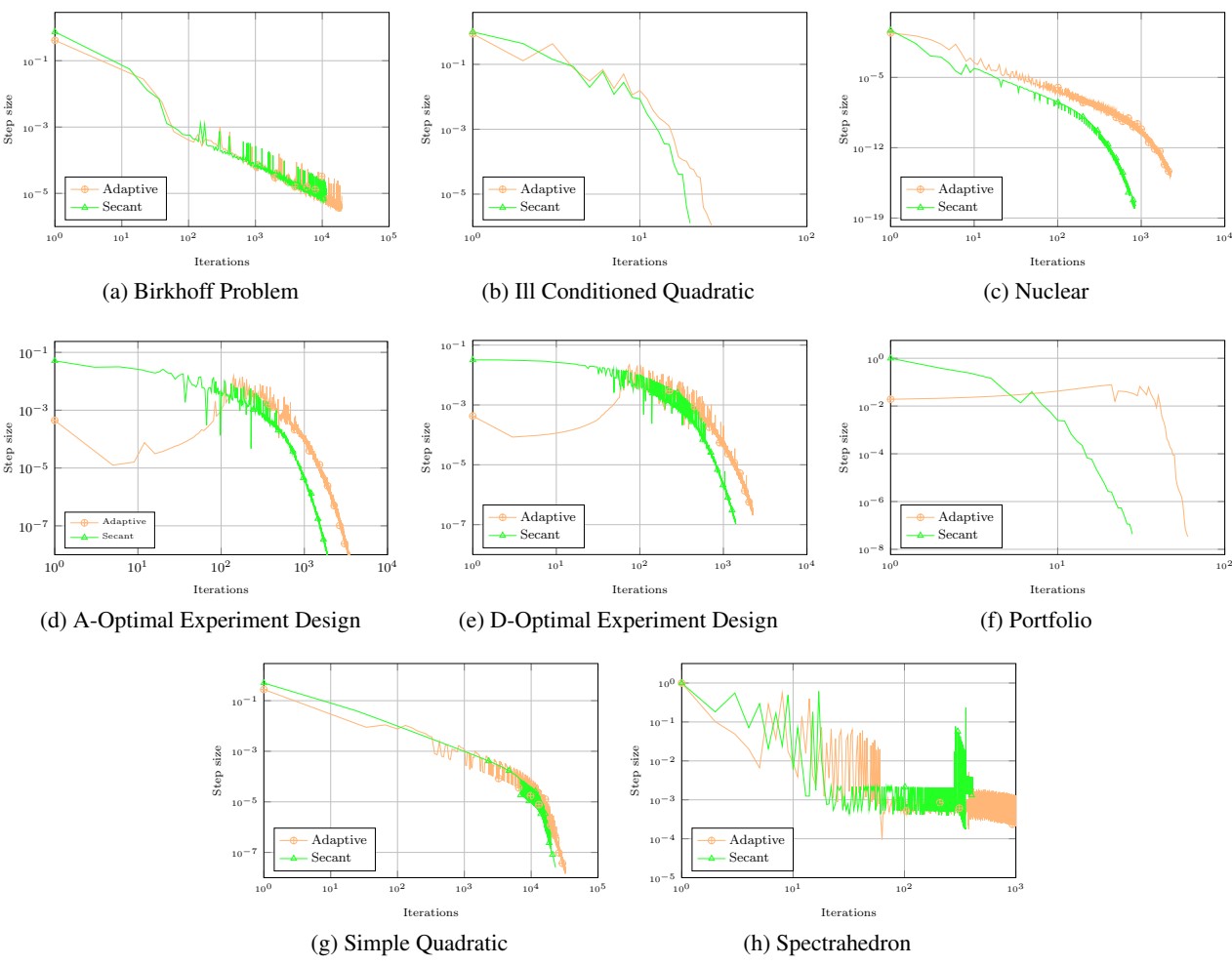

(a) Birkhoff Problem  (b) Ill Conditioned Quadratic  (c) Nuclear

(d) A-Optimal Experiment Design  (e) D-Optimal Experiment Design  (f) Portfolio

(g) Simple Quadratic  (h) Spectrahedron

*Figure 13.* Computed step sizes over iteration for the Secant line search and Adaptive line search on various problem classes.

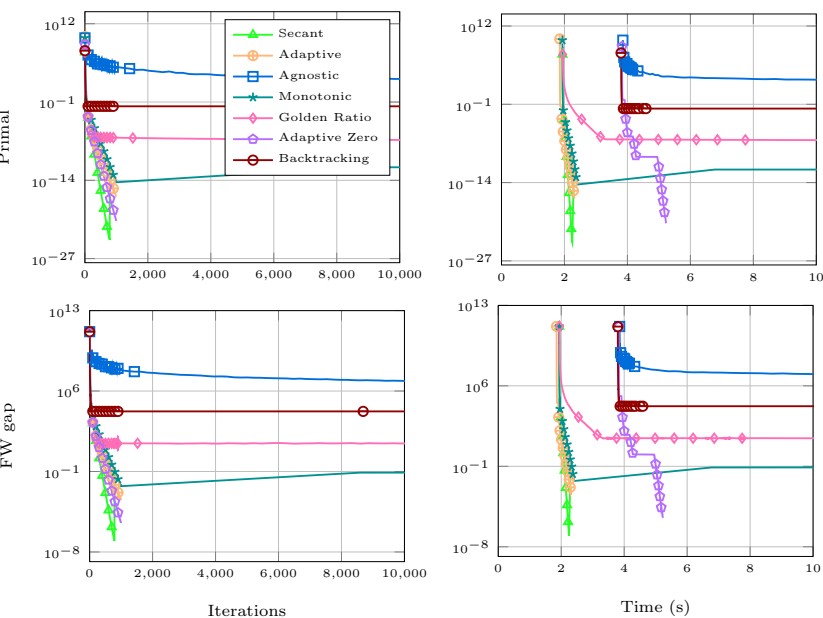

*Figure 14.* Progress of the primal value and FW gap for an instance of the Nuclear problem using Vanilla Frank-Wolfe.

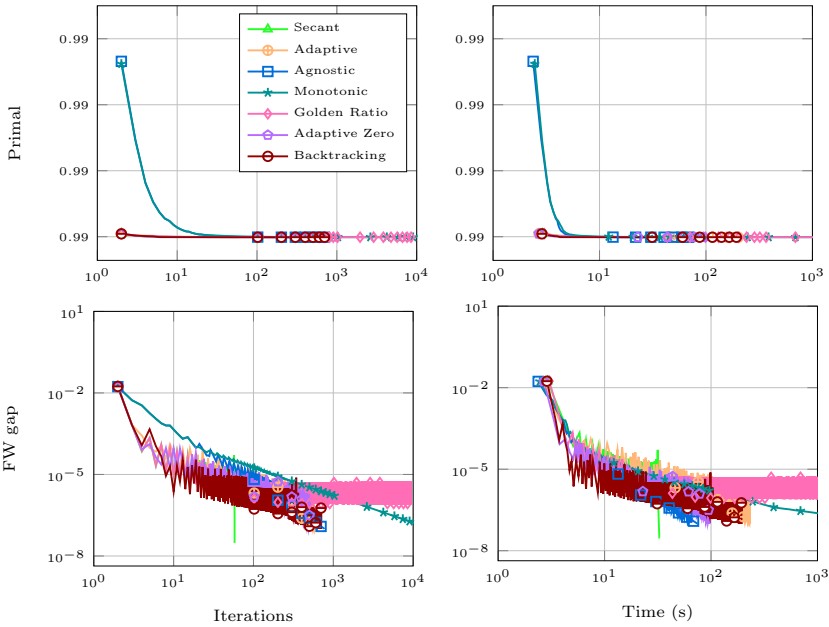

*Figure 15.* Progress of the primal value and FW gap for an instance of the Spectrahedron problem using Vanilla Frank-Wolfe.

