# OpenReview forum: "Secant Line Search for Frank-Wolfe Algorithms"
_ICML.cc/2025/Conference — ICML 2025 poster_

### Official Review · Reviewer_rqBT · 2025-03-11

**Overall Recommendation:** 2

**Summary:**

This paper introduces a new step-size strategy, the Secant Line Search (SLS), to optimize the Frank-Wolfe (FW) algorithm. SLS leverages the secant method to solve the line search problem, which reduces the computational cost compared to traditional methods. Theoretical guarantees for SLS’s convergence are provided, and numerical experiments show its superiority over other commonly used step-size strategies in terms of both computational performance and convergence speed.

**Claims And Evidence:**

The authors provide both theoretical analysis and experimental results to confirm their claim that SLS is computationally efficient and improves the convergence of Frank-Wolfe algorithms.

**Essential References Not Discussed:**

The authors do a good job of citing the relevant works in optimization and Frank-Wolfe algorithms.

**Experimental Designs Or Analyses:**

The experimental designs are sound, comparing SLS against other step-size strategies across a wide range of problems. The authors present results from several problem classes, demonstrating the practical applicability and efficiency of SLS.

**Methods And Evaluation Criteria:**

The proposed method, Secant Line Search (SLS), is a feasible approach to the Frank-Wolfe algorithm’s step-size problem. The evaluation criteria used for the experiments (such as the number of iterations and computational time) are appropriate and align with typical benchmarks in optimization problems.

**Other Comments Or Suggestions:**

N/A

**Other Strengths And Weaknesses:**

Strengths:
1. The proposed method addresses the step-size selection problem in a way that largely reduces the computational cost of the Frank-Wolfe algorithm while maintaining good convergence properties.
2. The experiments provide evidence of its effectiveness, particularly in terms of the number of iterations and overall computational time.

Weaknesses:

1. The paper lacks a detailed complexity analysis for the proposed methods, particularly in terms of sample complexity. For instance, how many gradient computations are required to achieve an $\epsilon$-optimal point? A more formal analysis would offer better insights into the theoretical performance of SLS.

2. In Lemma 2.1, the convergence guarantees rely on assumptions that seem quite strict, which may limit the broader applicability of the method.

3. The analysis in Section 2.2 raises some concerns, as the symbol "≈" is used frequently. This could suggest a lack of rigor in the analysis.

**Questions For Authors:**

See the Weakness part.

**Relation To Broader Scientific Literature:**

The key contributions of the paper relate well to the broader literature on optimization algorithms, particularly the Frank-Wolfe method.

**Theoretical Claims:**

Theoretical guarantees for the SLS strategy are provided and are supported by lemmas and theorems, including Lemma 2.1 and Theorem 3.1.

---

> ### Author Rebuttal · Authors · 2025-04-01
>
> We thank the reviewer for their feedback and questions.
>
> > The paper lacks a detailed complexity analysis for the proposed methods.
>
> We provide an analysis of the local superlinear convergence rate of the secant method and of its global convergence under suitable assumptions, which is an improvement over what can be stated in the unconstrained case, specifically relying on the setup of Frank-Wolfe algorithms in which the line search happens on a bounded line segment. This superlinear convergence guarantees that in practice, we perform an almost exact line search while remaining highly tractable, as can be noticed from the low number of secant iterations. We must highlight that our results are agnostic to the optimization problem, the precise convergence rate will depend on local properties of the function.
>
> > In Lemma 2.1, the convergence guarantees rely on assumptions that seem quite strict, which may limit the broader applicability of the method.
>
> The assumption might not be as strict as they seem; see Theorem 3.1. It suffices that the problem is strictly convex.
>
> > The analysis in Section 2.2 raises some concerns, as the symbol "≈" is used frequently. This could suggest a lack of rigor in the analysis.
>
> We use "≈" to ignore lower order terms for the sake of clarity. Specifically the contents of section 2.2 are provided for intuition, since it is folklore and the proof of this specific case is known and can be found, e.g., in Grinshpan (2024).
>
> To sum up, our main contributions are (A) we perform comprehensive experiments on the secant line search in a wide variety of benchmarks and against a wide variety of alternative step-sizes showing its advantages over previous approaches and (B) we observed that the structure of the FW algorithm is such that under mild assumptions it is always guaranteed to converge from any initialization (and the line search always occurs over a compact segment), which justifies that our method works in practice.

---

### Official Review · Reviewer_bzSe · 2025-03-12

**Overall Recommendation:** 3

**Summary:**

This paper proposes Secant Line Search (SLS),  a new step size strategy for Frank-Wolfe algorithms, by posing line search as root finding and using the secant method to solve it. The method is simple and easy to implement. The same principle can seemingly be applied in line search for algorithms other than Frank-Wolfe. The strategy is validated through numerical experiments.

**Claims And Evidence:**

As I understand it, the paper lists three contributions: proposing a new step size strategy based on the secant method, providing theoretical guarantees for this strategy when coupled with the Frank-Wolfe algorithm and demonstrating that the new strategy outperforms standard ones used for Frank-Wolfe.

Based on my assessment, I believe that only the last contribution may be substantiated.

I think that using the secant method for line search is not a new idea, if that was what the authors meant. For example, Chapter 7 in (E.K.P. Chong and S.H. Zak. An Introduction to Optimization. Fourth edition. Wiley, 2013) makes an explicit connection between line search and one-dimensional search methods such as the secant method. This connection is also mentioned in lectures notes such as https://www.princeton.edu/~aaa/Public/Teaching/ORF363_COS323/F14/ORF363_COS323_F14_Lec7.pdf
and some post at https://math.stackexchange.com/questions/3785724/line-search-using-secant-method
Please correct me if I misunderstood this but, to the best of my knowledge, this does not count as a contribution.

It also seems to me (see Theoretical Claims) that there may be an issue with the proof of lemma 2.1, which is the foundation of the further theoretical results. In any case, since SLS is not an exact line search, I think it would be important to establish a relationship between epsilon in the stopping criterion and the step size produced by SLS, and also its implications for convergence rates.

Finally, the paper does present empirical evidence that the method does work well in practice, but I think a more thorough discussion and clearer presentation of the results are required. First, in my view it would be much more informative if the results were grouped by problem in different subsections of the Computational Experiments section. Then, each problem could be presented in more detail, for example writing the objective function and/or describing the compact convex set on which each problem is defined as readers might not be familiar with typical benchmark problems for Frank-Wolfe. In this vein, it would be helpful to elaborate on what makes a good benchmark for Frank-Wolfe methods, which properties are stress-tested by which problem objectives, data and constraints. This would also help to delineate the problem setting in which SLS works best. Also, there could be an explicit mention of how the initial point of each problem instance was chosen and why, which I could not find in the paper. Finally, each subsection could contain a plot or a succinct table clearly conveying how SLS fares with respect to other methods.

**Essential References Not Discussed:**

As referenced above, for example in chapter 7 of (E.K.P. Chong and S.H. Zak. An Introduction to Optimization. Fourth edition. Wiley, 2013), using the secant method for line search does not seem to be a novel idea and it would be important to explain in detail what is novel in SLS with respect to previous work.

**Ethical Review Concerns:**

I wanted to flag a potential anonymity issue in the paper. On page 10, there’s a reference to an upcoming publication:

Wirth, E., Peña, J., and Pokutta, S. Accelerated affine-invariant convergence rates of the Frank-Wolfe algorithm with open-loop step-sizes. To appear in Mathematical Programming A, December 2024.

I am unsure if this violates ethical policies but I thought I should mention it.

**Experimental Designs Or Analyses:**

I checked the experiments and they look sound, but I think their presentation could improve by a lot, as remarked above.

**Methods And Evaluation Criteria:**

Yes, the evaluation criteria make sense, but I’m not sure how comprehensive the experiments are because I’m not very familiar with Frank-Wolfe literature

**Other Comments Or Suggestions:**

Minor:
The contribution paragraph "New step-size strategy" makes reference to some requirements that are described in the second paragraph of the Related Work section. Readers that skip the text before the contributions would not understand what the authors mean.
I would use a different font size for the contribution paragraphs and Preliminaries and Notation.
Recent approaches (e.g. https://arxiv.org/pdf/1905.09997) using stochastic line search in the context of neural networks also go by the name of SLS. FW and NN are probably sufficiently separated that this won’t cause any confusion, but I just wanted to let the authors know.
(Line 162, second column) The last sentence of the last paragraph of the second column on pg. 3 can be written more clearly, e.g., "x lies between the points a and y, and the differences delta(a,x) and delta(x,y)"
(Line 303, second column) The acronym BPCG is used here, but only defined later on line 706.
Making the plot colors darker (e.g. green and darkorange) in Figure 1 would make them easier to see.
Starting the discussion in section 4 with step size and iteration count remarks is distracting, I think the actual performance results should come first. That is, I think Table 1 and Figure 3 should come before Figures 1 and 2. First, you present empirical evidence that SLS works well through Table 1 and Figure 3 and then you could hypothesize why based on the step size and line search information conveyed by Figures 1 and 2.
Table 1 could have an extra "gain" column quantifying the performance gains/losses of SLS with respect to the best performing method among the remaining methods. Also, if a method is not able to reach the desired precision, it might be better to simply apply a different color to the performance metrics or highlight them somehow in that particular problem instance rather than reporting the dual gap, which is confusing. In particular, it seems that the results of the best performing method are reported in bold font, but shouldn’t these only include methods that actually reached the desired precision?
For problems in which the primal gap is not zero, I would suggest plotting primal-primal* in log scale instead of primal in linear scale, where primal* is the least primal gap found by all methods.

Typos:
(Line 64, first column) The three types of step-size strategies for FW have inconsistent numbering: (1), (ii), (iii)
Is gamma_{a}=gamma_{\ast}?
(Line 328, first column) Is the phrase “Nonetheless, the same analysis can be applied to these methods as well by carrying out over the.”” complete?

**Other Strengths And Weaknesses:**

The argument used in the proof of lemma 2.1 is nice, although I believe there is still some work to be done.

**Questions For Authors:**

1. On which problems SLS outperforms other line search methods?
2. What was the stopping criterion used in the experiments? For example, in Figure 3a Adaptive FW and SLS seem to stop at somewhat arbitrary points.
3. What makes a convincing benchmark for FW? What are the features that good experiments should capture?

**Relation To Broader Scientific Literature:**

The idea of using the secant method as a line search procedure could be more broadly applicable beyond Frank-Wolfe algorithms, but it seems that this idea is not new.

**Theoretical Claims:**

Lemma 2.1: don’t you have to show the ratio (S(x,y)-a)/(x-a) < 1-delta for some delta in (0,1)? Counterexample: if x_{n}=a+eps+(½)^{n} with small eps in (0,1), then 0< (x_{n+1}-a)/(x_{n}-a)=(eps+(½)^{n+1})/(eps+(½)^{n}) < 1, but x_{n} converges to a+eps>a. I don’t know if it’s possible to find some phi such that secant method produces the above x_{n+1}, but the point is that the condition that 0 < (S(x,y)-a)/(x-a) < 1 is not enough to prove that x_{n} and phi(x_{n}) converge to a and phi(a).
The local convergence analysis (section 2.2) assumes phi’(a) is nonzero, but SLS approximately finds precisely the point where the derivative of f(x_{t} - gamma * d_{t}) w.r.t. gamma is zero. Is it possible to overcome this assumption?
Since the critical point of f(x_{t} - gamma d_{t}) in terms of gamma is only solved approximately, it would be useful to understand the relationship between the approximation error epsilon and the actual convergence rate obtained with the SLS step size.

---

> ### Author Rebuttal · Authors · 2025-04-01
>
> We thank the reviewer for their feedback and questions on the paper.
>
> We agree that using root finding methods for line search is not new and in fact many line search methods, such as e.g., bisection line search is of that type. However, (Quasi-)Newton methods, secant methods, etc are traditionally *not* used as line search methods because in general there is no guarantee that they are globally converging. Put differently, *if* they converge they are fast but they may not converge at all. This is different here where we actually proved that in the setting of FW methods, where the step size is guaranteed to be confined between 0 and 1 (or 0 and a weight smaller than one for some FW variants), we *can* guarantee global convergence under mild assumptions that are often satisfied in contexts where FW methods are applied: we only require strict convexity.
>
> In Lemma 2.1, we do not show convergence by just saying that this property holds:  $ 0 < S(x,y) < 1 $. As it can be read at the end of our proof, we have instead that by $ x_{n+1} = S(x_n, x_{n-1}) $ and by that property then $ x_n -x_{n+1} $ converges to $0$. And because it is by definition $ \phi(x_n) / \Delta(x_n, x_{n-1}) = x_n - x_{n+1} \to 0 $ and $\Delta(x_n, x_{n-1})$ is upper bounded by a constant, it can only be that $ \phi(x_n) \to 0 $ and that can only be if $x_n \to a$, as specified in our proof.
>
> Question 3: We present each problem class in more detail, including the type of constraint set in the appendix due to space constraint. Our benchmark problems contain standard problems from the Frank-Wolfe literature, e.g. similar to some classes studied in Pedregosa et al 2018, Dvurechensky et al 2023, Carderera et al 2021 which all introduce new step-size strategies for FW. We were also careful to include different function classes (self-concordant, quadratics) and conditioning to be able to showcase the performance of SLS in a wide array of applications.
> Furthermore, our experiments assess multiple criteria for the proposed step size:
> 1. Is it making FW converge fast, as counted by the number of iterations? This question will typically have the same answer for all line searches given infinite precision (e.g. using arbitrary-size floating point numbers or rational arithmetic). However, when using 64-bit floating points, we showed that great differences can occur in behavior because of floating-point error accumulation, see Figure 5 in the appendix, in which golden ratio, another line search, quickly stalls in terms of FW gap precisely because of numerics. In contrast, we empirically observe the numerical robustness of SLS, even in ill-conditioned problems (when the Lipschitz constant is very high).
> 2. Is the method also effective in time, meaning that the good performance of the step size is not outweighed by the computational cost of the inner iterations evaluating the gradient at the candidate point?
>
> We answer both questions positively thanks to the computational experiments, evaluating the methods against iterations and time. We also compute the number of inner line search iterations throughout the Frank-Wolfe runs, empirically showing this remains consistently low throughout experiments. This ensures that SLS is also a suitable step size for problems where the gradient is particularly costly to evaluate (relative to the LMO).
>
> Q1: SLS shows superior performance on all problems with a quadratic objective function. It, in particular, shows stable performance even for ill-conditioned problems.
> Additionally, we also see a good performance on generalized self concordant objectives like the A-Optimal and D-Optimal Design problems and the Portfolio problem with log revenue.
>
> Q2: Frank-Wolfe is stopped if we reach either the dual gap of 1e-7 or a time of 1 hour (we will state this explicitly in the paper). The different strategies lead to different trajectories of Frank-Wolfe hence the different end points. Since there is a time limit, the dual gap value is of interest for the unsolved instances as we can compare how much progress each method obtained within the time. In the table, the geometric mean of the dual gap is only computed for the unsolved instances.
>
> The tolerance in the line search corresponds to one on the univariate derivative, which directly leads to a bounded function error ($\gamma \in [0,1]$). Others, e.g. Pedregosa et al 2020, Bomze et al 2019 have studied approximate LS for FW. We will highlight these references in the revised version.
>
> To sum up, our main contributions are (A) we perform comprehensive experiments on the secant line search in a wide variety of benchmarks and against a wide variety of alternative step-sizes showing its advantages over previous approaches and (B) we observed that the structure of the FW algorithm is such that under mild assumptions it is always guaranteed to converge from any initialization (and the line search always occurs over a compact segment), which justifies that our method works in practice.

---

### Official Review · Reviewer_dXfc · 2025-03-14

**Overall Recommendation:** 1

**Summary:**

The paper introduces a novel step-size strategy for Frank-Wolfe (FW) algorithms called Secant Line Search (SLS), which utilizes the secant method to determine step sizes efficiently.  SLS requires only function and gradient evaluations, making it computationally less expensive while adapting to the local smoothness of the function. The authors establish theoretical guarantees for convergence. Empirical results across various constrained optimization problems show that SLS achieves faster convergence and reduced computational in general.

**Claims And Evidence:**

Please refer to the "Other Strengths And Weaknesses" section.

**Essential References Not Discussed:**

Please refer to the "Other Strengths And Weaknesses" section.

**Experimental Designs Or Analyses:**

Please refer to the "Other Strengths And Weaknesses" section.

**Methods And Evaluation Criteria:**

Please refer to the "Other Strengths And Weaknesses" section.

**Other Comments Or Suggestions:**

1. The paper would be easier to understand if the related work section were moved after the introduction of the secant method.

2. Lines 40–48 are difficult to follow because they rely on techniques defined later in the paper.

3. The self-concordant property is not defined.

**Other Strengths And Weaknesses:**

Strength:

1. Empirical results demonstrate the effectiveness of the proposed line search method.


Weakness:

1. Limited theoretical novelty. The convergence guarantee is derived by directly combining the existing convergence results of the Frank-Wolfe method and the secant method.


2. Claims are not supported by sufficient evidence. It would be more convincing if references about the convergence of the Frank-Wolfe method were added in Theorem 3.1. Additionally, the ``linear convergence'' mentioned in lines 287–288 and the ``superlinear convergence'' in Remark 3.2 are not supported by proofs or references.

**Questions For Authors:**

None

**Relation To Broader Scientific Literature:**

Not sure.

**Theoretical Claims:**

Please refer to the "Other Strengths And Weaknesses" section.

---

> ### Author Rebuttal · Authors · 2025-04-01
>
> We thank the reviewer for their feedback and questions.
>
> > Limited theoretical novelty. The convergence guarantee is derived by directly combining the existing convergence results of the Frank-Wolfe method and the secant method.
>
> The theoretical novelty is indeed not the central element here, however in contrast to gradient descent methods where there it is hard (if not next to impossible) to make the secant method work consistently as a line search, in the context of Frank-Wolfe the Secant Line Search actually becomes a valid line search method. So the contribution here is really showing that in the settings were FW methods are usually applied we have a global theoretical guarantee that the secant method works.
>
> > Evidence of claims (more convincing if references about the convergence of the Frank-Wolfe method were added in Theorem 3.1). (the linear convergence mentioned in lines 287–288 and the superlinear convergence in Remark 3.2 are not supported by proofs or references)
>
>  We did not include an in-depth discussion about the local convergence of the secant method because it is not the main focus of the paper and has been established elsewhere. In fact, we cited Díez 2003 who provided an in-depth analysis and provided the convergence rate inducing equation $\lambda^m + \lambda^{m-1} = 1$, where m is the multiplicity of the root; $\lambda$ is then the rate we can expect. What we did provide though is a proof for the *global convergence* in common FW settings, which is the key crux for the secant method as it is not a globally convergent method in general. This way we can ensure that the secant method can be used as line search method.
> We will restate the Theorem etc to make this more clear.
>
> > Self concordance
>
> The self-concordant property is defined in (3.5), we will rephrase to make this more clear and will reference this in the introduction.
>
> To sum up, our main contributions are (A) we perform comprehensive experiments on the secant line search in a wide variety of benchmarks and against a wide variety of alternative step-sizes showing its advantages over previous approaches and (B) we observed that the structure of the FW algorithm is such that under mild assumptions it is always guaranteed to converge from any initialization (and the line search always occurs over a compact segment), which justifies that our method works in practice.

---

> > ### Comment · Reviewer_dXfc · 2025-04-06
> >
> > I appreciate the authors' rebuttal and will keep my current score.

---

### Official Review · Reviewer_vWCY · 2025-03-16

**Overall Recommendation:** 1

**Summary:**

This paper suggests that in Frank-Wolfe algorithm, one can use the Secant Method to set the step size for performance improvement.

**Claims And Evidence:**

Line 159, left column: $S(x,y) = S(y,x)$ is not true.

Proof of lemma 2.1 is dubious: by what monotonicity can one claim that $\frac{\Delta(x,a)}{\Delta(x,y)} > 0$? Also, the argument for $<1$ part is also shaky.

**Essential References Not Discussed:**

N/A

**Experimental Designs Or Analyses:**

N/A

**Methods And Evaluation Criteria:**

N/A

**Other Comments Or Suggestions:**

N/A

**Other Strengths And Weaknesses:**

N/A

**Questions For Authors:**

N/A

**Relation To Broader Scientific Literature:**

N/A

**Theoretical Claims:**

N/A

---

> ### Author Rebuttal · Authors · 2025-04-01
>
> > Line 159, left column: $S(x,y) = S(y,x)$ is not true.
>
> $ S(x, y) = S(y, x) $ holds. Writing the definition and multiplying by $\Delta(x, y) = \Delta(y, x)$ and reorganizing, we obtain that the expression is equivalent to $ (x-y)\Delta(x,y) = \phi(x) - \phi(y) $, which clearly holds true by definition of $ \Delta(x, y) $
>
> > Proof of lemma 2.1 is dubious: by what monotonicity can one claim that $\frac{\Delta(x,a)}{\Delta(x,y)} > 0$? Also, the argument for $<1$ part is also shaky.
>
> The monotonicity of $\phi$, as assumed in the statement, trivially implies that $\Delta(z, w) > 0$, for all $z, w \in \mathcal{U}$ as by definition monotonicity means $\phi(z) - \phi(w) > 0$ if $z - w > 0$.
> Our proof is correct, see also the response to Reviewer bzSe.

---

### Decision · Program_Chairs · 2025-05-01

**Decision:**

Accept (poster)

**Comment:**

The main contribution of this paper is to show that the secant method enjoys global convergence guarantees for the line-search step of the Frank-Wolfe method. The method is easy to implement and efficient, as validated through numerical experiments. While some reviewers raised limited theoretical novelty as a concern, I do not consider it as substantially limited; I find it interesting that a simple and overlooked mechanism can, in fact, provide an effective solution for the line search of the Frank-Wolfe algorithm. On the negative side, the analysis relies on a strict convexity assumption which limits its applicability and relevance. I strongly recommend the authors to follow the suggestions raised by the reviewers to further improve the presentation quality.